# Comprehensive CRISPR/Cas9-based mutagenesis identifies single-amino acid substitutions that abrogate SPEN function in X inactivation

Corinne Kaufmann , Sarah Sting, Chao Dai & Anton Wutz ✉

While genetic screens have facilitated the dissection of protein function in animal development, advances in systematic point mutagenesis open new opportunities for forward genetics in mammalian cells. Here, we develop a CRISPR/Cas9-mediated base editing screen that allows functional screening of extensive collections of single amino acid substitutions of endogenous proteins. We demonstrate the application on the X-chromosomal *Hprt* and the autosomal *Msh2* gene in diploid male and haploid mouse embryonic stem cells, respectively. Finally, we use this methodology to generate a sequence-function map of the transcriptional co-repressor SPEN in X chromosome inactivation. We demonstrate that the substitution of the SPEN RRM4-residue W522 abrogates X-linked gene repression by Xist RNA and impairs the establishment of H3K27me3 deposition. Our results demonstrate that screening in haploid cells allows efficient identification of mutations that would be recessive in diploid cells, suggesting applications across a wide range of areas.

In mammals, X chromosome inactivation (XCI) compensates for gene dosage differences arising from heteromorphic sex chromosomes[1,2]. The mechanism of developmentally regulated inactivation of one of the two X chromosomes has been extensively studied in mice and in mouse embryonic stem cells (ESCs) when they enter differentiation[3]. To initiate gene silencing, the long non-coding RNA (lncRNA) Xist is transcribed from and associates with the future inactive X chromosome (Xi). Repeat sequence motifs of Xist RNA have been shown to be required for chromatin changes and gene repression on the X chromosome. Chromatin modifications are mediated by Xist repeats B and C, which recruit Polycomb group proteins[4,5]. Ubiquitination of histone H2A lysine-119 (H2AK119ub1) by a non-canonical Polycomb repressive complex 1 (PRC1) containing PCGF3/5 is thought to establish an initial mark that mediates further PRC1 recruitment and H2AK119ub1 modification, as well as PRC2 recruitment via JARID2, leading to tri-methylation of histone 3 lysine-27 (H3K27me3)[6,7]. Multiple interactions amplify both chromatin modifications and are thought to support the spreading and substantial enrichment of Polycomb proteins across the Xi. X-chromosomal gene silencing is mediated by Xist repeat A. A mutation of repeat A abrogates X-chromosomal gene repression but does not affect chromatin changes on the Xi, including histone modifications and Polycomb complex recruitment[8,9]. SPEN, a 360 kDa multi-functional protein, has been identified as an Xist repeat A-interacting protein and an essential silencing factor for XCI[10–16]. N-terminal RNA-recognition motifs (RRMs) were demonstrated to interact with Xist repeat A in vitro[13,17] and in cells[10,18]. A SPOC domain at the C-terminus of SPEN has been shown to interact with the nuclear co-repressors 1/2 (NCoR/SMRT), which engage the histone deacetylase 3 (HDAC3)[19–21]. HDAC3 has been shown to be required for XCI[3], and a mutation of HDAC3 in ESCs delays gene silencing by Xist RNA[3]. A large intrinsically disordered region (IDR) separates the RRM and SPOC domains of SPEN and is required for its accumulation on the Xi[15,16]. The SPEN IDR is among the largest in the mammalian proteome. Although it remains largely unannotated, interaction with nuclear hormone receptors and the DNA-binding factor RBP-Jκ has been mapped to the IDR[22].

Institute of Molecular Health Sciences, Department of Biology, ETH Zurich, Zurich, Switzerland. ✉e-mail: awutz@ethz.ch

Mutation or depletion of SPEN in mouse ESCs has demonstrated its requirement for X-chromosomal gene silencing. Loss of SPEN has also been observed to decrease[13] or abolish Polycomb recruitment by Xist RNA[11]. This is surprising, as deletion of Xist repeat A has little effect on Polycomb recruitment and the establishment of associated histone modifications on the Xi[23,24]. Although the differential effect between the loss of SPEN and repeat A remains unexplained, it might indicate independent functions of SPEN in regulating Polycomb complexes. Homozygous mutations of SPEN in mice have been observed to cause defects in development and Notch signaling[25,26]. However, no sex-specific phenotypes have been reported. Lethality of a *Spen* mutation at E12.5[25] is observed later than that of an *Xist* mutation in female embryos, which is reported before E8.5[27]. Recently, a requirement for SPEN in imprinted XCI has been shown in preimplantation embryos[14]. The differing requirements of SPEN for random XCI in ESCs and mouse embryos remain to be defined and require a better understanding of the relationship between the different *Spen* mutations on the level of the full-length protein or its functional regions.

Functional exploration of SPEN has been challenging due to its large size and diverse cellular functions[18,19,22,28], and methodologies for efficient screening of autosomal genes at the endogenous expression level are missing. The expression level is often critical for developmental regulators. The fact that mutations of the endogenous *Spen* locus are recessive has impeded systematic analysis. In haploid cells, mutations are hemizygous and thus the problem of masked phenotypes of heterozygous mutations is solved. Screens in haploid cells – including one that identified SPEN as a silencing factor in XCI[13] – have aimed at efficient loss-of-function mutations using highly mutagenic gene trap vectors. However, null-mutations or complete loss-of-function alleles can limit advances in mechanistic understanding or lead to divergent conclusions when pleiotropic phenotypes are involved. Thus, chemically induced point mutagenesis, for instance, with N-Ethyl-N-Nitrosourea (ENU), has been performed in haploid ESCs and included exome sequencing that required clonal analysis[29]. Inducing point mutations allows the identification of individual amino acid residues involved in protein function and thus provides mechanistic insights. However, assignment of genotype-phenotype relationships is challenging in pooled chemical screens, and reliable discrimination of driver from passenger mutations requires stringent selection that is often not achieved for developmental pathways[30,31]. Recent developments in sequence-targeted base editors have opened new opportunities for mutagenesis screening. Tiling screens using nuclease-deficient Cas9 (dCas9) proteins to target deaminases have been deployed in diploid cells. Such base editors have been successfully used for identifying dominant mutations that confer resistance to clinically relevant inhibitors of oncogenic pathways[32,33]. Base editor screens can be applied to large cell pools without the requirement for resource-intensive clonal analysis. However, screening diploid cells is not suitable for identifying recessive mutations, as heterozygosity can mask or obscure phenotypic effects.

Here, we employ an engineered CRISPR/Cas9-mediated base editor to screen haploid ESCs to generate a comprehensive functional map at single amino acid resolution of a protein-of-interest. We demonstrate the utility of our haploid screening system by applying it to the autosomally encoded silencing factor SPEN in the context of its role in XCI. The successful identification of a loss-of-function mutation of SPEN, W522C, was verified by the functional analysis of a reconstituted diploid ESC line. The mutation of W522 impaired X-chromosomal gene silencing and the establishment of H3K27me3. Additionally, we demonstrate reduced enrichment of W522-mutated SPEN at the Xi upon *Xist* induction.

## Results

### Establishing targeted base editor-mediated point mutagenesis for the functional screening of HPRT

To develop a method for targeting point mutagenesis of endogenous genes, we adapted a hyperactive cytosine deaminase AID*Δ, which was previously derived from the activation-induced deaminase (AID) by deleting a C-terminal nuclear export signal and introducing three substitutions to enhance activity and nuclear localization[34]. We fused AID*Δ to the N-terminus of a catalytically-dead Cas9 (dCas9; D10A, H840A) to facilitate direct targeting by sgRNAs (Fig. 1a). dCas9 retains its helicase activity and leads to the formation of single-stranded DNA (ssDNA), which can be deaminated by AID*Δ. Additionally, dCas9 was fused to two uracil glycosylase inhibitors (UGIs) to prevent reversion of mutations by cellular DNA repair[35].

To assess the screen performance, we screened the X-chromosomal *Hprt* gene, which is hemizygous in male ESCs. Functional HPRT leads to the incorporation of 6-thioguanine (6TG) into the genome, leading to cell death. Thus, HPRT loss-of-function mutations can efficiently be selected by 6TG treatment. 23 sgRNAs were designed to target the coding region of *Hprt* with a spacing of approximately 50 bp to a maximum of 100 bp between sgRNAs (Fig. 1b). The base editing window of AID*Δ was estimated previously to extend approximately 50 bp upstream and downstream of the PAM site[34]. Expression vectors for the base editor and sgRNAs were co-transfected into male ESCs. Following transient selection for puromycin resistance encoded on the sgRNA expression vector, mutation profiles were determined before and after 6TG selection using a next-generation sequencing strategy. 6TG has previously been shown to be efficient for recovering HPRT mutations in ESCs with little to no background[29,36,37]. For determining mutation profiles, RNA was extracted from unselected and 6TG-selected cell pools and analyzed by targeted amplicon sequencing of the coding region of Hprt. cDNA-based amplicon sequencing reduces mutations that are not translated, such as mutations affecting splicing or introducing premature stop-codons, the latter of which are efficiently targeted by nonsense-mediated decay.

After preprocessing and alignment of sequencing reads, duplicates were removed to avoid overestimation of mutations. Base changes were then identified for each read and translated to the corresponding amino acid substitutions. Point mutations in the unselected cell pool were predominant for transitions of C > T and G > A, consistent with previous observations[34], confirming that most mutations resulted from targeted deamination by AID*Δ (Fig. 1c). Efficient and specific selection pressure on HPRT was indicated by a significant enrichment of C > T (rate ratio 9.7, FDR(5%)-adjusted $p = 0.013$) and G > A (rate ratio 7.9, FDR(5%)-adjusted $p = 0.002$) changes, as well as a significant enrichment of nonsynonymous mutations upon 6TG selection (two-way ANOVA with Sidak multiple comparisons $p = 0.002$) (Fig. 1d). Mean mutation frequencies (number of mutations relative to sequencing coverage) per amino acid residue (Fig. 1e) and mutation frequencies for each amino acid substitution possible were calculated per mutagenized cell pool (Fig. 1f). For the mutation frequency analysis, reads containing multiple amino acid substitutions were removed to avoid ambiguous assignment. Comparing the mutation frequencies between unselected cell pools and wild-type control cells confirmed that the entire coding region was targeted, with substitutions obtained for 196 out of 219 amino acids (89.50%) across a total of 8 cell pools. Individual cell pools were obtained from a starting population of approximately 100,000 cells before transfection and contained, on average, substitutions of 109 out of the 219 amino acids (Fig. 1g). The top 10 enriched amino acid substitutions per pool (frequency$_{selected}$/frequency$_{unselected}$) that showed a high mutation frequency in the selected population were considered screen hits and compared to reported mutations in studies and databases. Premature stop codons were excluded from the screen hits since truncations are not informative for the structure-function relationship of a protein. Human and mouse HPRT share an identity of 96.79% (ClustalOmega alignment of UniProt sequences P00493 (mouse) and P00492 (human)), which facilitated the comparison of our screening results with databases of human mutations. Using our cut-off, we selected loss-of-function substitutions of 26 amino acids, of which 30.77% were

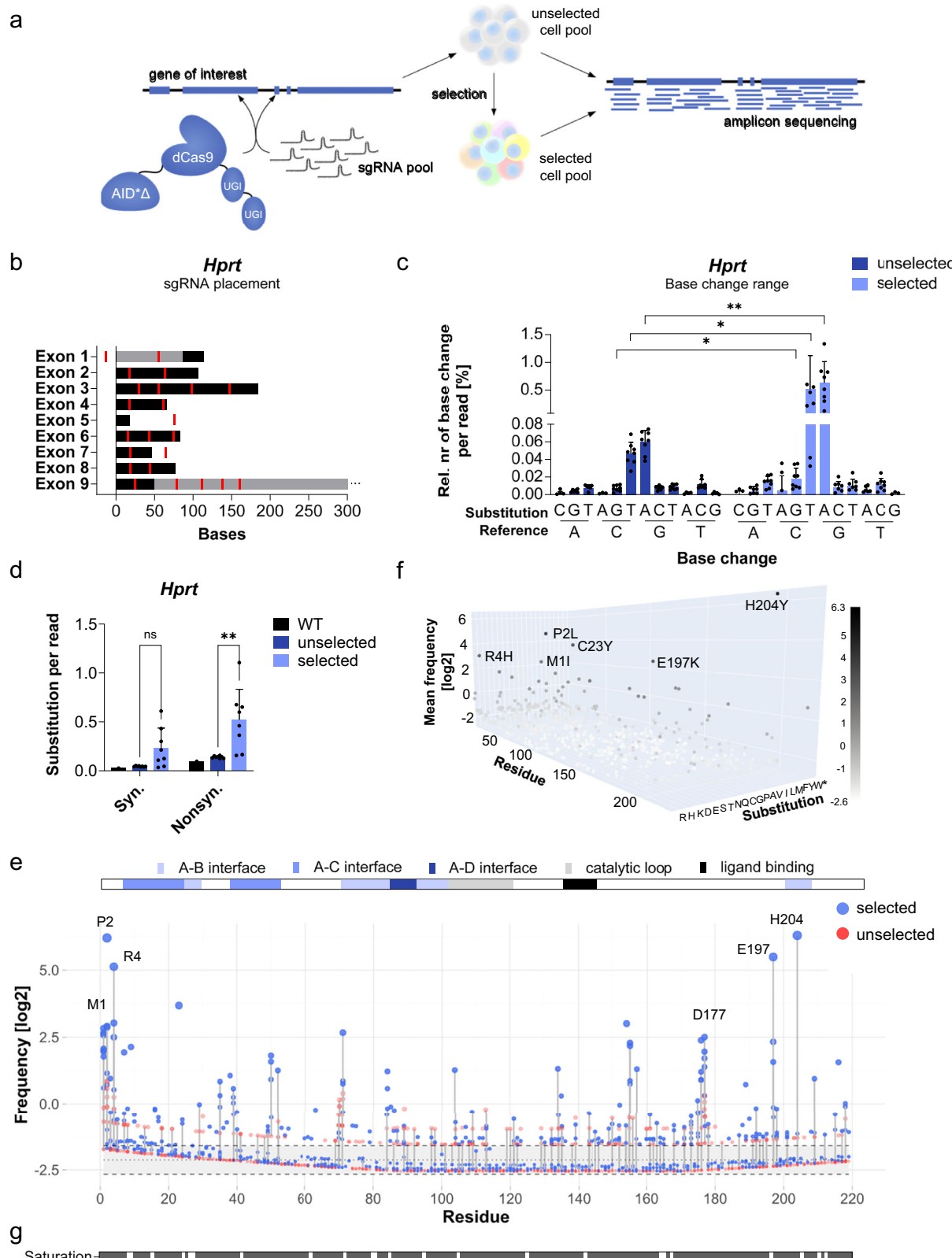

**a**

**b** *Hprt* sgRNA placement

**c** *Hprt* Base change range — unselected / selected

**d** *Hprt*

**f**

**e**

**g**

reported in ClinVar, 65.38% in Human Gene Mutation Database (HGMD), and 92.31% in a previous study[38] (Table 1). The latter summarizes mutations reported in the Human HPRT Mutation Data Base[39]. Among our screening hits, G26D and H204Y have been associated with Lesch-Nyhan syndrome. H204Y, which is located at the ligand binding site, was reported as a hotspot mutation in humans[38]. Therefore, our targeted deaminase base editor is capable of inducing substitutions with broad coverage, and our screening approach is sensitive and specific for identifying known disease mutations in HPRT.

**Base editor-mediated point mutagenesis for the functional screening of the autosomal *Msh2* gene in haploid mouse ESCs**

In diploid cells, mutations of autosomal genes are heterozygous, leading to ambiguity and masking of phenotypes. To extend screening

**Fig. 1 | Base-editing screen on *Hprt* in diploid mouse ESCs. a** Overview of the screening strategy. A hyperactive activation-induced deaminase (AID*Δ) was fused to a catalytically-dead Cas9 (dCas9) and two uracil glycosylase inhibitors (UGI). Mouse ESCs were co-transfected with the base-editing construct and an sgRNA pool targeting the Hprt coding region, followed by selection for loss-of-function mutations. Unselected and selected cell pools were analyzed by long-range PCR-based sequencing. **b** gRNA placement along the exons of *Hprt*. sgRNAs (red), and 5′ and 3′-UTR (grey) are visualized. **c** Base change frequencies in unselected and selected cell pools ($n = 8$ paired pools). Base change counts were normalized to the total read number to obtain base change frequencies. The frequencies were background-corrected using wild-type base change frequencies. G > A ($q = 0.002$ **), C > T ($q = 0.013$ *), and C > G ($q = 0.030$ *) changes were significantly enriched upon selection, as assessed by a paired quasi-Poisson generalized linear model on raw mutation counts, with read number as an offset. P-values were adjusted for multiple testing using the Benjamini-Hochberg method. Means and standard deviations are visualized as bars and error bars, respectively.

**d** Frequencies of synonymous and nonsynonymous mutations. Mutation counts were normalized to the total read number per sample (wild-type $n = 1$; unselected and selected $n = 8$ paired pools). Nonsynonymous ($p = 0.002$ **), but not synonymous ($p = 0.123$ ns), substitutions were significantly enriched upon selection, as assessed using a two-way ANOVA with a Sidak multiple comparisons test. Means are visualized as bars. Standard deviations are indicated with error bars. **e** Mutation frequencies shown in $\log_2$-scale for unselected (red) and selected (blue) cell pools ($n = 8$ paired pools). Mutation frequencies were calculated by normalizing the mutation count to the sequencing coverage of the given residue. Annotated protein domains are highlighted above the lollipop-plot[62]. Positions of the highest-ranked substitution per pool are labeled. **f** 3D-scatter plot of the mean frequencies for each amino acid substitution in selected cell pools. Frequencies are plotted in $\log_2$-scale. Substitutions with a $\log_2$ mean frequency ≥2 are labeled. $n = 8$ **g** Mutation saturation along all cell pools ($n = 8$). Residues were classified as mutated (grey) when their frequency exceeded that in the unselected wild-type pool.

**Table 1 | Summarized *Hprt* screen hits obtained from a combined analysis of high-frequency and simultaneously highly enriched amino acid substitutions**

| Substitution | Domain | Reported in Duan et al. 2004 Position | Reported in HGMD Substitution phenotype | Reported in ClinVar Position | Substitution phenotype | Position | Substitution | Note |
|---|---|---|---|---|---|---|---|---|
| M1I*/L* | START codon | x | | x | Lesch-Nyhan | x | | |
| P2L | | x | | | | | | |
| T3R/S | | x | | | | | | |
| R4H* | | | | | | | | |
| S7N | A-C | x | | x | | | | Glycine in humans |
| V9M | A-C | x | | | | | | |
| G16D | A-C | x | Lesch-Nyhan syndrome | x | Lesch-Nyhan syndrome | x | | |
| C23Y | A-C | x | | x | | | | |
| V35I | | x | | | | | | |
| P38F | A-C | x | | x | | | | |
| H39Y | A-C | x | | x | | | | |
| G40V | A-C | x | 6-TG resistance | x | | | | |
| A50V* | A-C | x | | x | | x | | |
| D52N | | x | | x | | | | |
| G71D | A-B | x | Multiple phenotypes | x | Lesch-Nyhan syndrome | x | | |
| A84T* | A-D/A-B | | | | | | | |
| S104N | catalytic loop | x | | x | | x | | |
| D135L | ligand binding | x | | x | | | | |
| S154N | | x | | | | | | |
| P155L*/H | | x | | x | | | | |
| M157I | | x | | | | | | |
| P176S* | | x | | x | | Hyperuricaemia | x | |
| V189L | | x | | x | | Lesch-Nyhan syndrome | x | |
| E197K* | A-B interface | x | | x | | | | |
| H204Y | A-B interface | x | Lesh-Nyhan syndrome | x | Lesch-Nyhan syndrome | x | Partial HPRT deficiency, Lesch-nyhan syndrome | Hotspot in humans (Duan et al., 2004) |
| S209N | | x | | | | | | |

| Duan et al. (2004) | | HGMD | | ClinVar | |
|---|---|---|---|---|---|
| **Positions** | **Substitutions** | **Positions** | **Substitutions** | **Positions** | **Substitutions** |
| 92.31% | 15.38% | 65.38% | 23.08% | 30.77% | 3.85% |

Substitutions that were observed in multiple samples are highlighted with an asterisk (*) in the substitution column. Hits were compared to Duan et al. (2004)[38], Human Gene Mutation Database (HGMD), and ClinVar, respectively. If the hit residue was listed in the corresponding database, it is indicated as a cross in the column 'Position'. If the same substitution of this residue was observed to cause a phenotype, it is mentioned under 'Substitution phenotype'. The lower table summarizes the relative number of positions or substitutions of the total number of screen hits covered by the corresponding database.

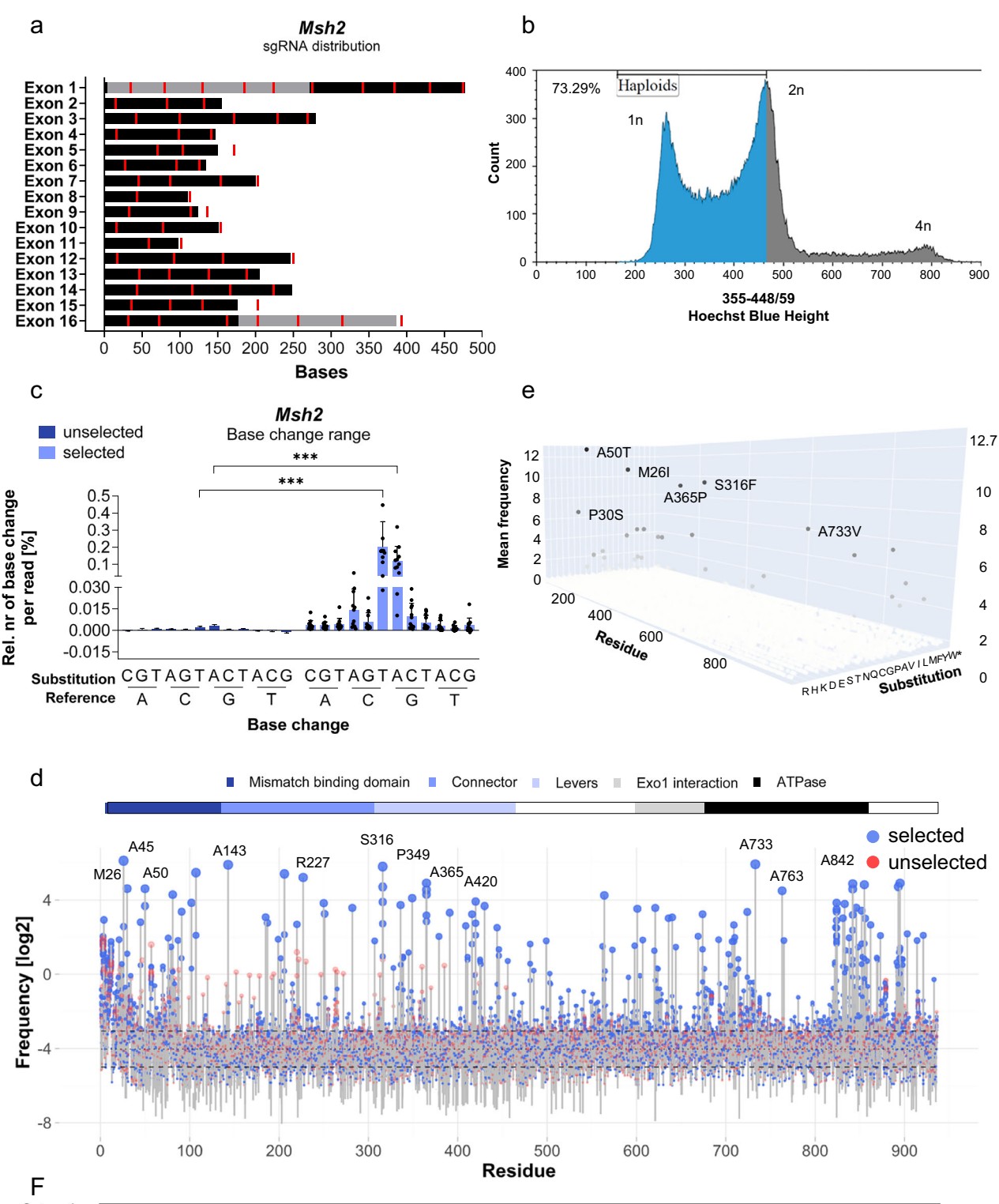

beyond dominant mutations, we adapted our targeted mutagenesis approach to haploid ESCs. The DNA mismatch repair (MMR) component MSH2 recognizes incorporated 6TG and triggers apoptosis[40,41], enabling the use of the same robust selection strategy as in our *Hprt* screen. Notably, 6TG has previously been used for the selection of haploid ESCs[29,42].

64 sgRNAs were designed to cover the coding region of Msh2 with a spacing of approximately 50 bp to a maximum of 100 bp between sgRNAs (Fig. 2a). Since haploid ESCs tend to diploidize in culture, cells were regularly enriched by FACS sorting after Hoechst 33342 staining

to select for G0/G1-phase haploid cells. Before mutagenesis, sorted pools were expanded twice to increase cell numbers, typically resulting in a cell pool containing ~70-75% haploid ESCs (representative Hoechst profile in Fig. 2b). To reduce the accumulation of spontaneous mutations during culture, we used haploid ESCs up to passage 20 for mutagenesis. After transient transfection and puromycin-selection, cDNA amplicon sequencing analysis of the Msh2 coding region confirmed strong enrichment of C > T (rate ratio 31.3, FDR(5%)-adjusted $p = 1.1*10^{-6}$) and G > A (rate ratio 12.9, FDR(5%)-adjusted $p = 1.1*10^{-5}$) transitions by 6TG selection, suggesting that most selected mutations

**Fig. 2 | Base-editing screen on *Msh2* in haploid mouse ESCs. a** gRNA placement along the exons of *Msh2*. sgRNAs (red), and 5′ and 3′-UTR (grey) are visualized. **b** Representative Hoechst 33342-staining DNA-profile of haploid ESCs used for mutagenesis screen. Two passages after sorting for culture expansion, the haploid ESC population contained approximately 70-75% of haploid ESCs. **c** Base change frequencies in unselected and selected cell pools (*n* = 12 paired pools). Base change counts were normalized to the total read number to obtain base change frequencies. The frequencies were background-corrected using wild-type base change frequencies. All base-change classes were significantly enriched upon selection (FDR(5%)-corrected *p* < 0.01 for all classes). The strongest effects were observed for C > T and G > A substitutions (rate ratios 31.326 and 12.878, respectively). Statistical significance was assessed using a paired quasi-Poisson generalized linear model on raw mutation counts, with read number as an offset. P-values were adjusted for multiple testing using the Benjamini-Hochberg method. Means and standard deviations are visualized as bars and error bars, respectively. **d** Mutation frequencies shown in log$_2$-scale for unselected (red) and selected (blue) cell pools (*n* = 12 paired pools). Mutation frequencies were calculated by normalizing the mutation count to the sequencing coverage of the given residue. Annotated protein domains are highlighted above the lollipop-plot[63]. Positions of the highest-ranked substitution per pool are labeled. **e** 3D-scatter plot of the mean frequencies for each amino acid substitution in selected cell pools. Frequencies are plotted in log$_2$-scale. Substitutions with a log$_2$ mean frequency ≥6 are labeled. *n* = 12 **f** Mutation saturation along all cell pools (*n* = 12). Residues were classified as mutated (grey) when their frequency exceeded that in the unselected wild-type pool.

were the result of targeted deamination (Fig. 2c). This observation is comparable to our *Hprt* screen in diploid ESCs. We identified loss-of-function substitutions at 54 positions over 12 individual cell pools.

Looking at mean substitution frequencies per residue in MSH2 (Fig. 2d), it appears that overall, there is less tolerance for substitutions compared to HPRT, since residues with a high mutation frequency in selected cell pools were spread across the whole protein. This observation was expected, since the majority of MSH2 consists of functional domains. However, looking at the mean frequency of specific substitutions (Fig. 2e), it appears that there is a tolerance to certain amino acid substitutions, but specific changes strongly impaired MSH2 function in DNA-repair. For instance, S316F and A365P are both located in α-helices from the lever domain, which undergoes conformational changes upon mismatch binding[43]. These mutations were observed to be enriched in multiple selected cell pools, indicating that the specific substitutions strongly impact residue function.

Mutation saturation was similar to our *Hprt* screen and overall, 865 of 936 (92.41%) amino acids were mutated, with an average of 474.63 residues per cell pool (Fig. 2f). Since human and mouse MSH2 share 92.5% identity (ClustalOmega alignment of UniProt sequences P43247 (mouse) and P43246 (human)), enriched amino acid substitutions from our screen were compared to ClinVar, HGMD, and a recent study that performed a mutagenesis screen with plasmid encoded MSH2 mutants transfected into a cell line with an *Msh2* frameshift mutation in exon 6[44]. Our screen hits contained 59 substitutions in a total of 54 residue positions. 92.59%, 35.19%, and 22.22% of the identified positions overlapped with ClinVar, HGMD, and Jia et al. (2021), respectively (Table 2). Although substitutions identified in our *Msh2* screen overlapped with known disease-associated mutations at a comparable rate to our *Hprt* screen, we note a lower overlap with the large number of predictions of Jia et al. (2021). This could have several reasons, including different cell systems, thresholds for predicted phenotypic substitutions, and the specificity and sensitivity of the screening setups. Based on overlap with known disease mutations, our results compare favorably with earlier work.

The overlap of predicted candidates with reported mutations in ClinVar shows that our screening approach can successfully identify mutations in an endogenous autosomal gene.

## Base editor-guided point mutagenesis screening of the large multi-domain protein SPEN reveals mutational flexibility and functional relevance of individual amino acid residues in XCI

We next applied our haploid screening approach to understanding the function of SPEN as a silencing factor in XCI. To facilitate selection, we used a haploid ESC line with a Doxycycline (Dox) inducible Xist allele (TX)[45] (Fig. 3a). *Xist* induction in haploid TX ESCs results in the initiation of XCI and silencing of the X chromosome, which results in drastically reduced cell survival. We had previously used this selection strategy successfully to recover null-mutations of *Spen*[13]. After 4 days of Xist induction, we observed cell death in approximately 90% of TX compared to wild-type ESCs, indicating considerably higher background than in our 6TG screens, where near complete selection was achieved. To cover the whole coding region of *Spen*, we designed 132 sgRNAs with a spacing of approximately 50 bp to a maximum of 100 bp between sgRNAs (Fig. 3b). Haploid TX cells were transfected with gRNA and AID*Δ expression vectors, then split into a control and a Dox-treated cell pool. After RNA extraction, the *Spen* coding region was amplified in six overlapping PCR reactions and sequenced. Analysis of 13 replicates showed that amino acid substitutions were distributed across the full SPEN protein (Fig. 3c). We obtained mutations for 3,524 out of 3,644 amino acids (96.71%) of SPEN (Fig. 3d) and selected substitutions of 46 positions in the protein as potential loss-of-function candidates (Table 3).

We used computational tools including the conservation predictor Consurf[46], the phosphorylation site predictor MuSiteDeep[47], the intrinsic disorder predictor AIUPred[48], as well as a predictor for residues undergoing phase separation, ParSe2[49], for understanding potential functions of selected substitutions (Table 3). A screen hit, S1284, which is a residue of the IDR, had a phosphorylation-site prediction score of 90.7% and is the fifth highest potential phosphorylation site in SPEN according to MuSiteDeep. We experimentally confirmed phosphorylation of S1284 by phospho-proteomics (Fig. 3e). Notably, phosphorylation was detected only after *Xist* induction but not before, indicating a potential regulation during the initiation of XCI.

The RRMs revealed several amino acid substitutions – including E448, T450, A489, and L503 of RRM3, or W522 and T536 of RRM4 – that lead to a loss-of-function of SPEN. Among them, we identified a substitution of the residue W522 (W522C), which extends into a highly conserved interface between RRM3 and RRM4 and is located at one of five canonical RNA-binding sites of the β-sheet surface of RRM4[17].

Altogether, our screen identified amino acid substitutions within critical structural elements and functional domains of SPEN that are predicted to disrupt its function as a silencing factor in XCI. The predicted strong phenotypic impact and high precision of point mutations enable the detailed dissection of the molecular functions of SPEN.

## SPEN W522 is required for X-chromosomal gene silencing by *Xist*.
To validate our mutagenesis screen, we functionally characterized the effects of the W522C substitution in SPEN. For this, we introduced the W522C substitution into the endogenous *Spen* locus in female TX XGFP/Cast ESCs that had been established from a hybrid cross of mice harboring an inducible Xist allele (TX) and a GFP reporter on the X chromosome (XGFP) with inbred *Mus musculus castaneous*. The polymorphisms between the two subspecies facilitate allelic expression analysis from the TX chromosome. The W522C substitution was introduced using CRISPR/Cas9-mediated homology-directed repair (HDR). Additionally, a silent mutation of the PAM sequence was introduced to prevent repeated cleavage and enable genotyping. We obtained two ESC lines that each contained the W522C substitution on one allele, and a 1 bp insertion (W522C/Ins) or a three amino acid

**Table 2 | *Msh2* screen hits obtained from a combined analysis of high-frequency and simultaneously highly enriched amino acid substitutions are summarized in the table (top)**

| Substitution | Domain | Reported in Jia, 2021 Position | Reported in HGMD Deleterious substitution | Reported in ClinVar Position | Substitution phenotype | Position | Substitution |
|---|---|---|---|---|---|---|---|
| M26I | mismatch binding | | no data | | | x | |
| P30S | mismatch binding | | | | | x | |
| A45T | mismatch binding | x | | x | | x | Hereditary nonpolyposis colorectal neoplasms |
| A50T/V | mismatch binding | x | | | | x | A50T: LS, HCPS, HNCN; A50V: LS, HCPS, HNCN |
| S81I | mismatch binding | | | | | x | HCPS, HCNC |
| D91V | mismatch binding | | | | | x | LS, HCPS, HNCN, Muir-Torré syndrome |
| V102I | mismatch binding | | | x | Colorectal cancer, non-polyposis | x | LS, HCPS, HNCN |
| A107V* | mismatch binding | | x | x | | x | LS, HCPS |
| A143V | connector | | | | | x | HCPS |
| S185F | connector | | | | | x | LS, HCPS, HNCN, Muir-Torré syndrome |
| L187F | connector | x | | x | | x | HCPS, HNCN |
| T206I | connector | | | | | x | HCPS, HNCN |
| G220E | connector | | | x | | x | HNCN |
| R227K | connector | | | x | | x | |
| G250R | connector | | | | | x | HCPS, HNCN |
| E251K | connector | | no data | | | x | LS, HCPS, HNCN |
| D282E | connector | | | | | x | HNCN |
| V307I | levers | | | | | x | |
| S316F* | levers | | | | | x | HCPS, HNCN |
| A336T | levers | | | x | | | |
| P349S | levers | x | x | x | | x | LS, HCPS, HNCN |
| A365P*/V* | levers | | | | | x | A365V: HCPS, HNCN |
| D379N | levers | | no data | | | x | |
| A391T | levers | | | x | | x | LS, HCPS, HNCN |
| S416N | levers | | | | | x | |
| A420V | levers | | | | | x | HCPS, HNCN |
| A430S | levers | | | x | | | |
| R444H | levers | | | | | x | LS, HCPS, HNCN |
| L481V | clamp | | | | | x | |
| A499G | clamp | x | | | | x | |
| T564I | levers | | | x | | x | HNCN |
| V598I | levers | x | | | | x | |
| H601Q | levers | | | | | | |
| R621Q | ATPase | x | no data | x | Colorectal cancer, non-polyposis | x | LS, HCPS, HNCN |
| A636V | ATPase | x | | x | | x | LS, HCPS, HNCN |
| A640V | ATPase | | | | | x | |
| G669D | ATPase | x | x | x | Colorectal cancer, non-polyposis | x | LS, HCPS, HNCN |
| G692W | ATPase | x | no data | x | Colorectal cancer | x | LS, HCPS, HNCN |
| L709F/I | ATPase | | | | | x | |
| T724I | ATPase | | | x | | x | |
| A733V | ATPase | | | x | | x | |
| A763T | ATPase | | | x | | x | HCPS |
| A765P | ATPase | x | x | | | x | |
| D823N | ATPase | | no data | | | x | HCPS, HNCN |

**Table 2 (continued) | Msh2 screen hits obtained from a combined analysis of high-frequency and simultaneously highly enriched amino acid substitutions are summarized in the table (top)**

| Substitution | Domain | Reported in Jia, 2021 Position | Reported in HGMD Deleterious substitution | Reported in ClinVar Position | Substitution phenotype | Position | Substitution |
|---|---|---|---|---|---|---|---|
| A831D | ATPase | x | x | | | x | |
| L833I | ATPase | | | | | x | |
| R838K | ATPase | | | | | x | |
| A842V/T | ATPase | | | | | | |
| E852K | ATPase | | | | | x | HCPS, HNCN |
| N856K | ATPase | | | | | x | |
| G862E | ATPase | | no score | | | x | |
| R873H | ATPase | | | x | | x | |
| P895F/L | ATPase | | | | | x | P895L: LS, HCPS, HNCN |
| S921T | ATPase | | | | | x | |

| Jia et al. (2021) | | HGMD | | ClinVar | |
|---|---|---|---|---|---|
| Positions | Substitutions | Positions | Substitutions | Positions | Substitutions |
| 22.22% | 8.47% | 35.19% | 6.78% | 92.59% | 49.15% |

Substitutions that were observed in multiple samples are highlighted with an asterisk (*) in the substitution column. Hits were compared to Jia et al. (2021)[44], Human Gene Mutation Database (HGMD), and ClinVar, respectively. If the hit residue was listed in the corresponding database, it is indicated as a cross in the column 'Position'. If the same substitution of this residue was observed to cause a phenotype, it is mentioned under 'Substitution phenotype'. The lower table summarizes the relative number of positions or substitutions of the total number of screen hits covered by the corresponding database. *LS* Lynch syndrome, *HCPS* Hereditary cancer-predisposing syndrome, and *HNCN* Hereditary nonpolyposis colorectal neoplasms.

deletion of V521, W522, and L523 (W522C/Δ) on the second allele, respectively (Fig. 4a). Therefore, neither of the cell lines expresses a wild-type version of SPEN that contains the W522 residue. Due to the recessive nature of *Spen* mutations, we do not expect an effect of the reduced copy number in the W522/Ins cell line. As controls, we generated cell lines with homozygous deletions of the RRM regions of SPEN (SPEN-/-1 and SPEN-/-2).

To assess the effect of the mutations on X-linked gene repression by Xist RNA, we performed RNA-seq analysis. The polymorphism between TX XGFP and Cast X chromosomes can be used to assess allele-specific gene expression, where Doxycycline-mediated *Xist* induction specifically silences the X-linked genes on the TX chromosome, while the Cast alleles remain active. We observed that the TX allelic ratio (TX SNP frequency relative to sequencing coverage) was strongly decreased after *Xist* induction in parental TX XGFP/Cast ESCs (Fig. 4b). The TX allelic ratio only slightly decreased after *Xist* induction in W522C/Ins and remained unchanged in W522C/Δ, similar to the two SPEN-/- cell lines. To additionally assess X chromosomal gene silencing upon naturally induced XCI, we differentiated the wild-type as well as the two W522-mutant cell lines to epiblast-like cells (EpiLCs). Gene expression was analyzed after 72 h of differentiation (Fig. 4c). While in the wild-type cell line X chromosomal gene expression was reduced upon differentiation, both W522C/Ins and W522C/Δ exhibited defects in silencing X chromosomal genes (Fig. 4d). *Xist* expression was equally upregulated in all three cell lines upon differentiation (Fig. 4e). We noted that Xist RNA levels remained lower in both W522C/Ins and W522C/Δ upon Dox induction compared to the wild-type cell line. Nevertheless, Xist RNA levels were increased in W522C/Ins and W522C/Δ cells under both conditions without consequential X-chromosomal gene silencing.

To further investigate the silencing defect on the chromatin level, we analyzed acetylation (H3K27ac) and tri-methylation (H3K27me3) of histone H3 lysine 27 over X-linked genes. Deacetylation is one of the earliest events after *Xist* expression[3]. SPEN has been implicated in the deacetylation of H3K27ac through its interaction with NCoR and HDAC3[19–21]. To analyze H3K27ac, we performed CUT&RUN using an H3K27ac-specific antibody. Acetylation of autosomal promoters was unaffected by doxycycline-induced *Xist* expression in wild-type, W522-mutant, and SPEN-/- ESCs (Fig. 4f). While an approximately 50%

reduction in acetylation over X-linked gene promoters (of previously-defined active genes[50]) after *Xist* induction was observed in parental TX XGFP/Cast ESCs, no decrease was observed in the SPEN-/- 1 and SPEN-/- 2 cells. For the W522-mutant cells, we observed variable deacetylation efficiencies, indicating a residual function of SPEN. Previously, SPEN mutations have also been reported to affect the recruitment of Polycomb repressive complexes and establishment of associated histone modifications[11,13]. To detect H3K27me3, we performed native ChIP followed by high-throughput sequencing. We observed a substantial H3K27me3 increase over X-linked genes after *Xist* induction in wild-type cells, whereas no increase was observed in the SPEN-/- 1 and SPEN-/- 2 cells. In both cell lines carrying the W522-mutation, we observed a markedly reduced and variable increase in H3K27me3 compared to wild-type cells. Taken together, our data indicate an impairment of chromatin changes by Xist in W522C-mutant ESCs.

To understand the defect caused by the W522C mutation, we further investigated the behavior of Xist RNA and SPEN. First, we visualized and analyzed Xist RNA cluster area per nucleus after 4 d of doxycycline-induced *Xist* expression (Fig. 5a). For this, we performed RNA FISH, where we used a double-stranded probe that can also detect Tsix lncRNA, which is expressed from the active X chromosome and visible as small dots. For the quantification of Xist RNA clusters, we calculated the area of the signal from the Xist cluster in relation to the area of the entire nucleus (defined by DAPI staining). Xist RNA areas were similar between wild-type and the two W522C-mutant cell lines after 4 d of Dox induction (Fig. 5b). However, RNA-seq readouts showed lower Xist RNA levels, especially in W522C/Δ (Fig. 4e), which does not seem to prevent cluster formation in our system. This observation does not suggest a significant effect of the W522-mutation on Xist RNA stability as it has been observed in previous studies about SPEN mutations[51]. However, while Xist RNA cluster formation appears to be unaffected in W522C-mutant cells, our assay does not allow an assessment of Xist RNA distribution over subregions of the X chromosome.

To advance our mechanistic understanding of the SPEN W522-mutation, we considered that it might affect SPEN binding affinity for Xist RNA based on its position within RRM4. SPEN has been shown to be enriched over the Xi after *Xist* induction, which has been shown to be required for X-chromosomal gene silencing by Xist[14,15,52].

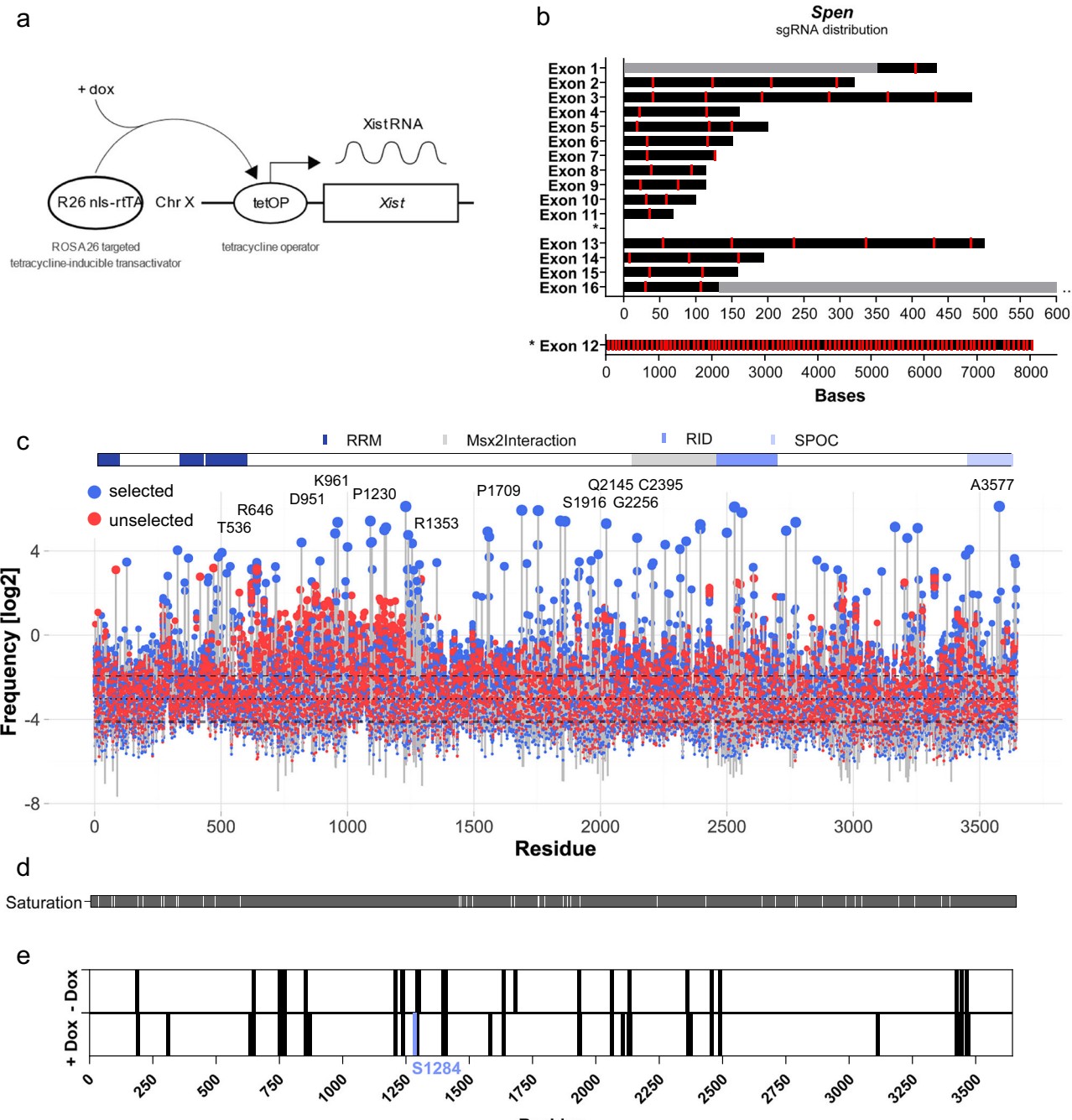

**Fig. 3 | Base-editing screen on *Spen* in haploid mouse ESCs. a** Schematic overview of the doxycycline-inducible *Xist* expression system in haploid TX XGFP ESCs used for the selection of SPEN loss-of-function mutations. A tetracycline-inducible trans-activator transgene is expressed from ROSA26 locus (R26 nls-rtTA), and upon doxycycline (dox) treatment activates the tetracycline operator (tetOP) leading to *Xist* expression. *Xist* expression-induced XCI in haploid ESCs leads to cell death in wild-type but not in mutant cells. **b** gRNA placement along the exons of *Spen*. sgRNAs (red), and 5′ and 3′-UTR (grey) are visualized. **c** Mutation frequencies shown in log₂-scale for unselected (red) and selected (blue) cell pools (*n* = 13 paired pools). Mutation frequencies were calculated by normalizing the mutation count to the sequencing coverage of the given residue. Annotated protein domains are highlighted above the lollipop-plot (Uniprot Q62504). Positions of the highest-ranked substitution per pool are labeled. **d** Mutation saturation along all cell pools (*n* = 13). Residues were classified as mutated (grey) when their frequency exceeded that in the unselected wild-type pool. **e** Phosphoproteomics by mass spectrometry on diploid TX XGFP/Cast hybrid ESCs. Phosphorylation sites along SPEN were assessed with and without doxycycline-induced *Xist* expression, respectively. S1284, which was identified as a potential loss-of-function site, was identified as a phosphoserine (light blue). *n* = 1.

To investigate this, we introduced an RFP-tag to the C-terminus at the endogenous *Spen* locus in wild-type and W522-mutant cells to study the localization of SPEN upon XCI induction. To allow live cell imaging in a system where *Xist* is equally expressed in all three cell lines according to the RNA-seq readout, we performed EpiLC differentiation and analyzed the RFP signal after 24, 48, and 72 h (Fig. 5c). In wild-type cells, focal SPEN-RFP signals were observed in 20% of the cells after 72 h of differentiation (Fig. 5c, d). Subsequent fixation and RNA FISH confirmed co-localization of the RFP signal with Xist RNA clusters. For both W522-mutant cells, SPEN clusters could be observed. However, the number of cells with focal RFP signal was drastically reduced (Fig. 5c, d).

**Table 3 | *Spen* screen hits obtained from a combined analysis of high-frequency and simultaneously highly enriched amino acid substitutions are summarized in the table**

| Substitution | Domain | Conservation (ConSurf) | Phosphorylation (MSiteDeep) | Intrinsic disorder (AIUPred) | Phase separation (ParSe2) |
|---|---|---|---|---|---|
| G127E | | 9 | | 0.9852 | P |
| L328F | | 7 | | 0.9475 | D |
| Q371R | RRM2 | 9 | | 0.2546 | F |
| E448K | RRM3 | 9 | | 0.2495 | F |
| T450M | RRM3 | 9 | 0.104 | 0.2376 | F |
| A489V | RRM3 | 9 | | 0.1081 | F |
| L503F | RRM3 | 9 | | 0.1582 | F |
| W522C | RRM4 | 8 | | 0.1558 | F |
| T536I* | RRM4 | 9 | 0.084 | 0.1142 | F |
| R646K | | 6 | | 0.9425 | P |
| A677I | | 9 | | 0.9842 | P |
| R818C | | 4 | | 0.9972 | D |
| P858S | | 9 | | 0.9989 | D |
| A891V | | 7 | | 0.9933 | F |
| D951N | | 4 | | 0.986 | D |
| K961N | | 4 | | 0.9824 | D |
| A1000V | | 9 | | 0.8002 | F |
| D1236N | | 9 | | 0.9938 | P |
| S1284F | | 4 | 0.907 | 0.9915 | P |
| P1353A | | 4 | | 0.986 | P |
| P1620S | | 7 | | 0.9986 | D |
| P1689S | | 3 | | 0.9993 | D |
| P1709F | | 3 | | 0.9996 | D |
| A1752V | | 8 | | 0.9989 | D |
| P1754S | | 6 | | 0.9991 | D |
| R1842C | | 6 | | 0.9953 | D |
| R1861K | | 9 | | 0.9846 | D |
| S1916N* | | 9 | 0.719 | 0.9968 | P |
| T1965I | | 7 | 0.658 | 0.9991 | D |
| Q2145H | MSX2/RID | 3 | | 0.9988 | D |
| G2256K | MSX2/RID | 3 | | 0.9989 | D |
| A2316V | MSX2/RID | 5 | | 0.9994 | P |
| C2395Y | MSX2/RID | 2 | | 0.9991 | D |
| V2559M | MSX2/RID | 9 | | 0.9526 | F |
| E2604Q | MSX2/RID | 8 | | 0.5775 | D |
| A2734T | RBPSUH | 3 | | 0.8173 | F |
| A2772V | RBPSUH | 9 | | 0.9298 | P |
| A2856V | | 9 | | 0.7859 | F |
| P2886S | | 9 | | 0.5971 | P |
| V2938I | | 9 | | 0.4306 | F |
| H2959C/Y | | 5 | | 0.8029 | F |
| G3112D | | 5 | | 0.9556 | D |
| V3165I | | 9 | | 0.7473 | F |
| P3459S | | 5 | | 0.9083 | D |
| A3577V | SPOC | 9 | | 0.1063 | F |
| M3637I | SPOC | 9 | | 0.1151 | F |

Substitutions that were observed in multiple samples are highlighted with an asterisk (*) in the substitution column. Prediction and computational tools were used to gather additional information on the screen hits, including: ConSurf[46], MuSiteDeep[47], AIUPred[48], and ParSe2[49].

We conclude that the mutation of W522 results in fewer cells containing clear SPEN accumulation after XCI induction, which is consistent with the idea that the mutation weakens the binding affinity of SPEN for Xist RNA. However, our data suggest that SPEN W522C can still be recruited to the X chromosome. As a consequence of inefficient recruitment of SPEN, we observe drastic abrogation of gene repression by Xist and reduced H3K27me3 deposition. The drastic impact of this single-amino acid mutation also validates the computational strategy of our screen.

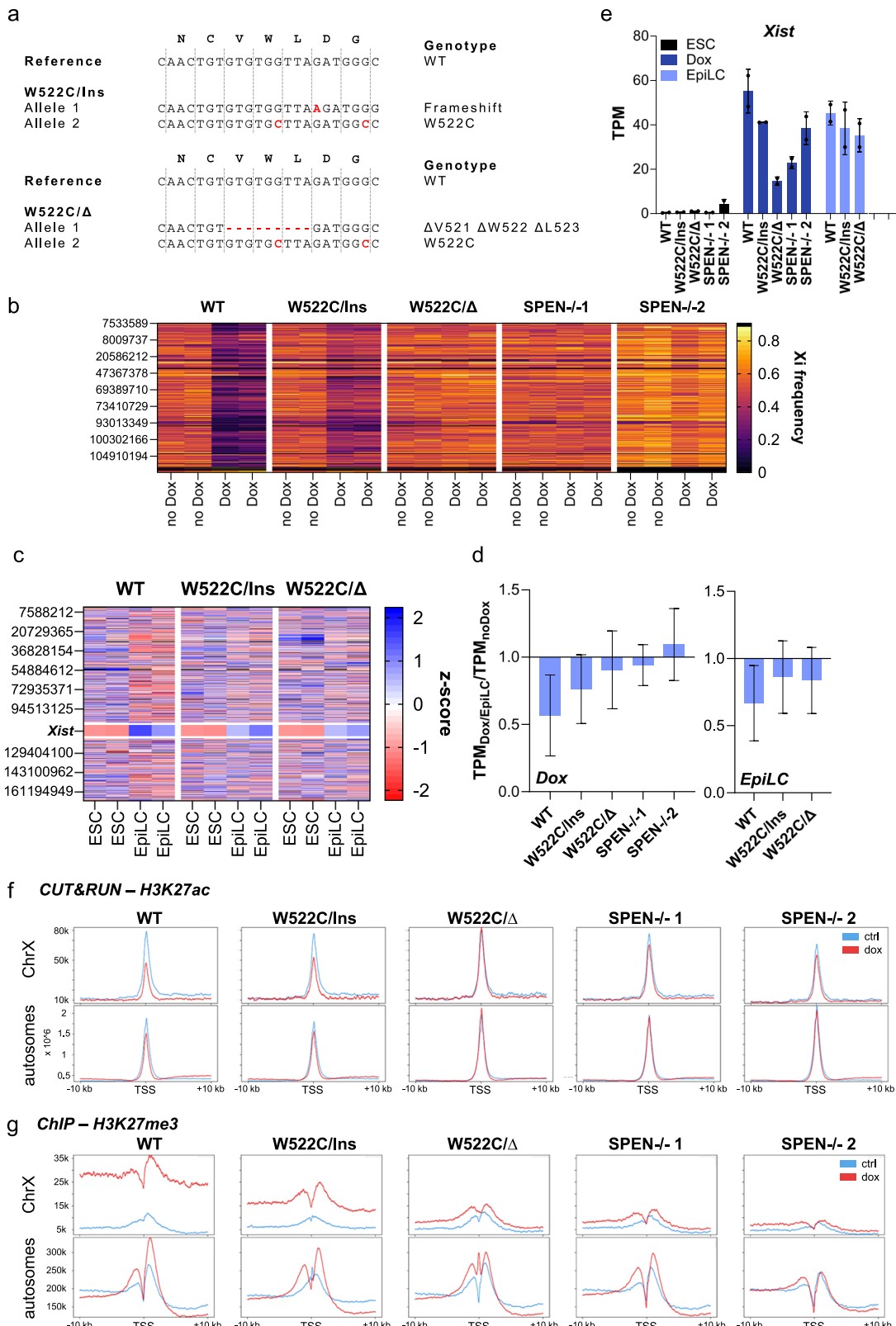

## Discussion

Our study profiles the function of SPEN in XCI at single-amino acid resolution and characterizes the effect of the substitution of W522 on the initiation of XCI. W522 is located at the canonical RNA-binding sites on the β-sheet surface of RRM4 and faces the intramolecular space between RRM3 and RRM4, which is involved in the interaction with a lncRNA in a published structure of SPEN[17]. The interaction between

RRM2-4 and Xist repeat A has been shown by mobility shift assays in vitro, and UV crosslinking RNP immunoprecipitation (CLIP) in cells[10,13,53]. Furthermore, RRM4 is required for Xist RNA interaction, and RRM3 enhances the binding affinity in vitro[54]. Our observation that SPEN W522C still localizes to the Xi after *Xist* induction indicates that Xist RNA binding is not entirely abrogated. However, we find a drastic reduction in the number of cells exhibiting SPEN accumulation

**Fig. 4 | Impact of the SPEN W522 mutation on Xist-mediated X-chromosomal gene silencing. a** Genotypes of SPEN W522 mutant cell lines. Clone W522C/Ins carries one W522C allele and one frameshift allele caused by a 1 bp insertion, whereas clone W522C/Δ contains one W522C allele and a 9 bp deletion removing V521, W522, and L523. **b** Allele-specific RNA-seq analysis of X-linked gene expression in wild-type, W522C-mutant, and SPEN-/- cell lines after 4 days of doxycycline-induced *Xist* expression. TX allele frequencies are shown for each single nucleotide polymorphism (SNP) ordered by its position along the X chromosome (y-axis). $n = 2$. **c** Quantitative analysis of gene expression by RNA-seq in wild-type and W522C-mutant cell lines differentiated to epiblast-like cells (EpiLCs) for 72 h. Gene expression was normalized to transcripts per million (TPM) values and z-scored within biological replicates. Chromosome coordinates are indicated in the heatmap. Missing values are shown in black. $n = 2$. **d** Quantification of global X-chromosomal gene silencing upon doxycycline-induced and differentiation-driven *Xist* expression. Mean TPM ratios ($n = 2$) relative to the naïve ESC state were calculated per gene, filtered to remove extreme values (top and bottom quartiles),

and averaged across genes. Under doxycycline-induced *Xist* expression (Dox), median log$_2$-ratio to wild-type cells increases progressively from W522C/Ins, W522C/Δ, SPEN-/- 1 and SPEN-/- 2, corresponding to 1.26x, 1.46x, 1.51x, and 1.68x increases, respectively. In differentiated cells (EpiLC), median log$_2$-ratio ratios from W522C/Ins and W522C/Δ were 1.18x and 1.20x, respectively. Mean ratios and standard deviations are visualized as bars and error bars, respectively. **e** Xist RNA expression levels measured by RNA-seq TPM values in ESCs, doxycycline-induced ESCs, and EpiLCs. $n = 2$. **f** CUT&RUN profiles of H3K27ac in wild-type TX XGFP/Cast, W522-mutant, and SPEN knockout cells with (red) and without (blue) *Xist* induction, plotted around transcription start sites (TSS) of active genes (known H3K4me3 sites[50]). The x-axis indicates distance to TSS. The y-axis corresponds to normalized read counts. **g** ChIP profiles of H3K27me3 in wild-type TX XGFP/Cast, W522-mutant, and SPEN knockout cells with (red) and without (blue) *Xist* induction, plotted around TSS of active genes (known H3K4me3 sites[50]). The x-axis indicates distance to TSS. The y-axis corresponds to normalized read counts.

compared to wild-type cells after XCI induction. This result is consistent with a lower binding affinity of the mutant SPEN RRMs to Xist-repeat A. The reduced efficiency of X-linked gene silencing by Xist in W522-mutant cells could be explained as a consequence of the reduced accumulation of SPEN on the Xi. We compare our W522-mutant cell lines to cells containing a CRISPR/Cas9-mediated deletion of the RRM regions. We observe evidence for gene silencing in the W522-mutant cell lines. In contrast, X chromosomal gene expression was unchanged after Xist induction in the RRM-mutant cells. The strong impairment of gene repression by Xist RNA in cells carrying the W522C mutation also reduces the establishment of H3K27me3 over X-linked genes. Wild-type cells show a substantial increase in H3K27me3 deposition over X-linked gene transcription units after XCI induction, whereas there is a complete lack of increase in H3K27me3 in RRM-mutant cells SPEN-/-1 and SPEN-/- 2.

In wild-type as well as W522-mutant cells, we observed Xist RNA cluster formation as visualized by RNA FISH. This would suggest that localization of Xist RNA is not impaired by the W522-mutation. This is in contrast to other SPEN loss-of-function mutations that have been investigated in previous studies, where the mutations resulted in reduced Xist RNA stability and lower Xist RNA levels[51,52]. We reconcile these different observations with the fact that the W522-mutant SPEN protein is still recruited by Xist, albeit to a lower amount.

We conclude that the isolation of the function of SPEN in XCI by the substitution of W522 provides a highly specific tool for further exploring this mechanism. Reduced pleiotropic effects of the W522 mutation raise the possibility of separating XCI and other functions of SPEN in development, which would be impossible to achieve by partial or complete deletion.

## Targeted base editor for mutagenesis screens in haploid ESCs
To screen SPEN, we have engineered a cytosine deaminase-dCas9 fusion protein for the introduction of point mutations and show that it can achieve broad mutation saturation in ESCs. To increase the mutation efficiency and randomness, several modifications to the previously reported AID*Δ deaminase[34] were made, including the fusion with dCas9 and two UGIs to facilitate sequence-specific targeting with gRNAs while reducing reversions by base-excision repair mechanisms. The new base editor extends the existing repertoire of tools for mutagenesis in ESCs. We demonstrate the application by identifying mutations in the endogenous X-chromosomal gene *Hprt*, or autosomal genes *Msh2* and *Spen* in either diploid or haploid ESCs, respectively. Screening in haploid ESCs has the advantage that it is not limited to dominant mutations. Previously, cDNA-based transgenes have also been used to assess recessive mutations. Although such screens can be performed in diploid cells, they require large collections of vectors for the expression of mutant cDNAs. Furthermore, expression of transgenes from heterologous promoters is often not

desirable, as overexpression poses risks for unknown effects on the phenotype. For large proteins, including SPEN, cDNA-based approaches are prohibitively difficult due to decreased delivery efficiency of large cDNA fragments.

Screening of point mutations in haploid ESCs has previously relied on chemical mutagenesis[30,31]. Chemically-induced mutations are distributed genome-wide, which can be advantageous depending on the screen design. However, mutation assignment in pooled screens requires stringent discrimination of driver from passenger mutations, which can only be partially solved by clonal analysis and whole exome sequencing[29]. The reduced sensitivity requires stringent selection, which can rarely be achieved for developmental systems. We demonstrate that our approach can cope with considerable background. By restricting the sequencing to regions that were targeted by the deaminase, the difficulty in recovering mutations and overall sequencing efforts can be reduced dramatically. The advantage of targeted mutagenesis is also reflected in the increasing application of base editors for screening of resistance mutations that have dominant phenotypes in diploid cells[32,33].

In addition to SPEN W522C, our screen has identified a set of single-amino acid substitutions that indicate interesting positions in SPEN. This is a resource for further exploration of the mechanism of XCI initiation. We identify S1284 as a phosphorylation site that is phosphorylated upon *Xist* induction in Doxycycline-treated ESCs. It will be interesting to explore S1284 phosphorylation in the regulation of XCI in the future. Interestingly, our screen did not identify R3531 and R3533 within the SPOC domain, which were reported to be involved in the interaction with the NCoR/SMRT and NuRD complexes in human cells. This observation indicates that the two residues likely have a redundant role rather than being essential as individual residues. However, it could also indicate that our screen has not reached saturation in this region. Future advances that could include a combination of cytosine and adenine deaminases might increase coverage and enable saturated mutagenesis. The potential for identification of a broad range of mutations, including loss-of-, gain-of-, and separation-of-function alleles, inspires applications of our screening approach in a wide range of areas, including human development and disease using human haploid ESCs[55].

## Methods
### Cell culture
**Diploid ESC culture.** Diploid ESCs were cultured in DMEM (high glucose) (Gibco) supplemented with 15% FBS (biowest), MEM NEAA (Gibco), sodium pyruvate (Gibco), 114.4 μM β-mercaptoethanol (Sigma-Aldrich), and 1000 U/mL mouse Leukemia Inhibitory Factor (LIF) (homemade). The medium was freshly supplemented with 3 μM CHIR99021 (Axon Medchem), 1 μM PD0325901 (Axon Medchem) (ESC +2iLIF). Cells were maintained at 37 °C and 5% CO$_2$. In this study, a male

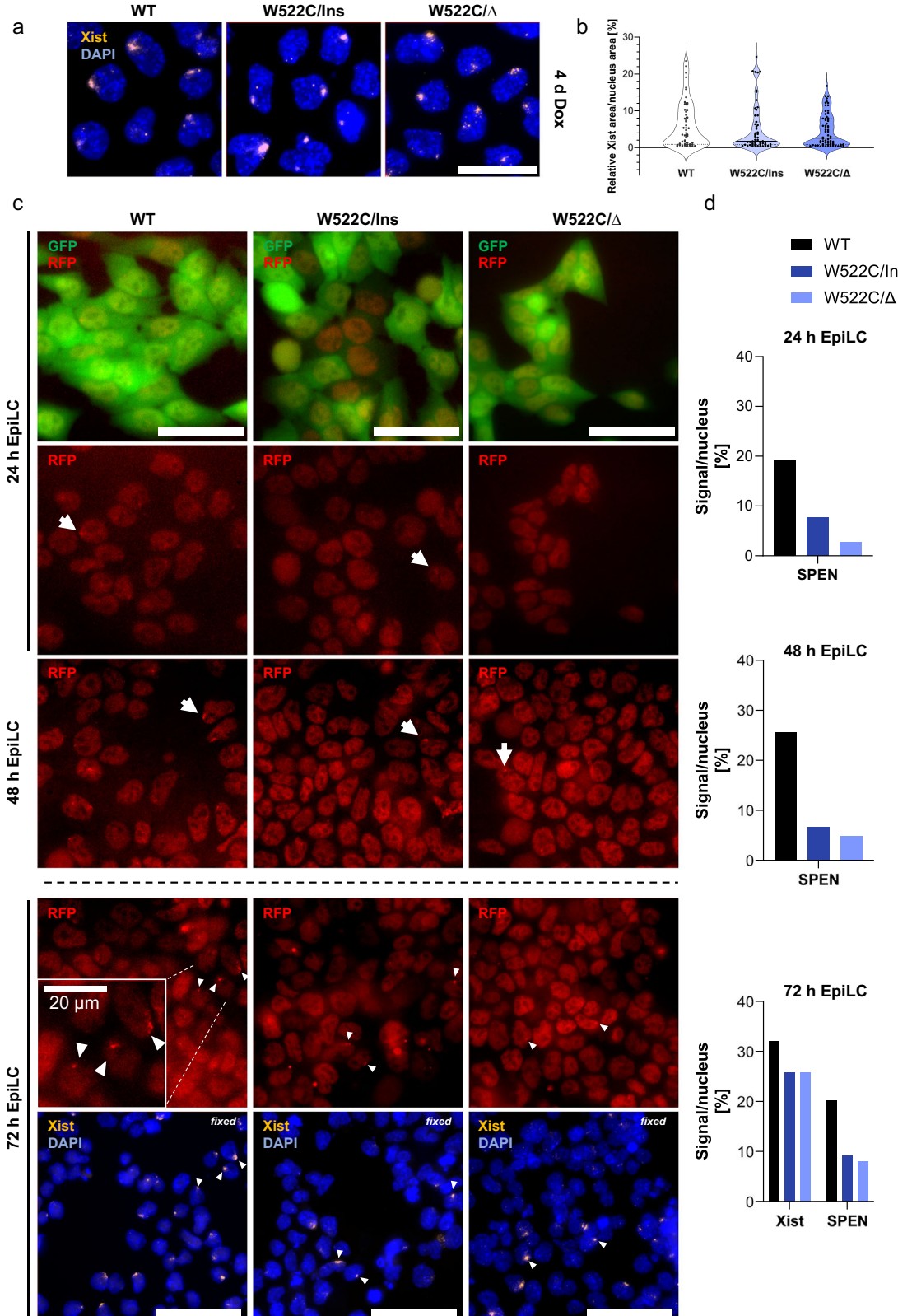

ESC cell line from C57bl/6 J&N mice (mutagenesis screen on *Hprt*) and a female TX XGFP/Cast cell line from TX XGFPxCast hybrid mice (*Mus musculus* x *Mus musculus castaneus*; a *Reverse tetracycline-controlled transactivator* (*rtTA*) at ROSA 26 locus (*Gt(ROSA)26Sor*, Chr6: 113067248-113077333), and a Tet Response Element (TRE) leading to the tetracycline-induced activation of *Xist*, and a *CAG-eGFP* (Tg(CAG-EGFP)50Osb) transgene on the X chromosome[56,57]) were used.

**Haploid ESC culture.** Haploid ESCs were cultured in NDiff 227 (Takara Bio) supplemented with 3 µM CHIR99021 (Axon Medchem), 1 µM PD0325901 (Axon Medchem), and 1250 U/mL LIF (homemade) (NDiff +2iLIF) on gelatine-coated plates. Haploid ESCs tend to diploidize during prolonged culture and thus were regularly purified by Fluorescence-Activated Cell Sorting (FACS). For that, cells were stained with Hoechst 33342 (Thermo Fisher Scientific) and subsequently, cells

**Fig. 5 | Impact of the SPEN W522 mutation on Xist RNA and SPEN localization.**
**a** Xist RNA clusters in both W522-mutant and wild-type ESCs after doxycycline-induced *Xist* expression. Representative RNA FISH images showing Xist RNA (yellow) clusters in wild-type and W522-mutant cells after four days of *Xist* induction by doxycycline. DNA is stained with DAPI (blue). Images are z-projections. scale bar = 25 μm. **b** RNA FISH image quantification of Xist cluster area relative to the nuclei area. Relative values were calculated for individual signals and nuclei and are plotted individually for wild-type (*n* = 49), W522C/Ins (*n* = 56), and W522C/Δ (*n* = 67). *n* ≥ 49 cells per sample. **c** Live cell imaging of endogenous SPEN-RFP during EpiLC differentiation. The RFP-tag was generated in TX XGFP/Cast cell lines, which are constitutively expressing *GFP* and exhibit a GFP signal in the nucleus and cytoplasm, which provided a control for the nuclear localization expected for SPEN-RFP. Representative RFP foci and accumulations are highlighted with white arrows. Images were taken after 24, 48, and 72 h of differentiation, respectively, and followed by an RNA FISH to visualize Xist RNA for the last time point. Co-localizing RFP foci and Xist clusters are visualized with white triangles for the 72 h timepoint. Images are z-projections. scale bar = 50 μm. **d** Quantification of the relative number of SPEN-RFP signals and/or Xist clusters per nucleus. Cell nuclei counting was done blinded; all intact and distinguishable nuclei were counted. For SPEN-RFP, all clear foci or accumulations that were distinguishable from the background signal were counted. For Xist RNA clusters, all signals, including dispersed dots and clusters were counted. For 24 h EpiLC: n ≥ 62 nuclei per sample. For 48 h, EpiLC: *n* ≥ 212 nuclei per sample. For 72 h EpiLC: *n* ≥ 213 nuclei per sample.

of the haploid G0/G1-peak were sorted. Cells were maintained at 37 °C and 5% $CO_2$. For this study, the haploid cell line TX XGFP was derived from parthenogenetic blastocysts obtained from TX XGFP mice, containing a *Reverse tetracycline-controlled transactivator* (*rtTA*) at ROSA 26 locus (*Gt(ROSA)26Sor*, Chr6: 113067248-113077333), a Tet Response Element (TRE) leading to the tetracycline-induced activation of *Xist*, and a *CAG-eGFP* (Tg(CAG-EGFP)50Osb) transgene on the X chromosome[42,56–58].

Cells were regularly tested for the absence of mycoplasma contamination. Diploid ESC stocks were additionally karyotyped to ensure the presence of two structurally normal X chromosomes. Cell pools that had undergone fewer than 25 passages were used for the screens to avoid the accumulation of spontaneous mutations.

**EpiLC differentiation.** To differentiate ESCs into Epiblast-like cells (EpiLCs), plates were coated with human fibronectin (16.6 μL/mL in PBS) (Corning) for at least one hour at 37 °C. ESCs were harvested and directly plated in freshly prepared EpiLC medium consisting of N2B27 (495 mL of DMEM/F12 (Thermo Fisher Scientific) and 480 mL of Neurobasal (Thermo Fisher Scientific) medium supplemented with 10 mL B27 supplement (Thermo Fisher Scientific), 5 mL N-2 supplement (Thermo Fisher Scientific), 2 mM L-glutamine (Thermo Fisher Scientific), 0.1 mM β-mercaptoethanol (Sigma-Aldrich), and penicillin-streptomycin (Thermo Fisher Scientific)) freshly supplemented with 10 μL/mL knock-out serum replacement (KSR; Thermo Fisher Scientific), 20 ng/mL Activin A (Qkine), and 12 ng/mL Fibroblast growth factor 2 (bFGF; Qkine). Medium was replaced daily with freshly made EpiLC medium until processing of the cells.

**Doxycycline induced *Xist* expression.** To induce *Xist* expression, cells were treated with 1 μg/mL Doxycycline hyclate (Dox) (Sigma-Aldrich) supplemented into ESC + LIF medium (without 2i supplementation). Treatment was done for four days, except when it was used for a mutation selection (See "Screen selection"). Medium was freshly made and changed daily.

**Screen selection**
**6TG selection.** Mutant cells in *Hprt* and *Msh2* screens were selected for loss-of-function mutations by treating the cells with 2 μM and 1.5 μM 6TG (Sigma-Aldrich), respectively, freshly supplemented into the corresponding culture medium. To allow mutation establishment in the cells, selection started after 6–7 days and was continued for 10 and 7 days, respectively, until no surviving cells were observed in the wild-type population and no excessive cell death had been observed in the mutagenized cell pool for 2–3 days.

**Doxycycline-mediated *Xist* induction.** Mutant cells in the *Spen* screen were selected for loss-of-function mutations by treating the cells with 1 μg/mL Doxycycline hyclate (Dox) (Sigma-Aldrich) supplemented into ESC + LIF medium (without 2i supplementation). Selection started 5 days post-transfection to allow the establishment of mutations in the cell pool and was continued for about 11 days until no excessive cell

death had been observed in the wild-type and mutagenized cell pool for 4–5 days.

**Cloning**
All plasmids obtained by cloning were validated for their correct sequence by Sanger sequencing and by size assessments.

**AIDhyp-dCas9-2xUGI.** Cloning with pGH335_MS2_AID*Δ-Hygro (insert; Addgene #85406) and BE4 (Addgene #100802) to replace APOBEC1 with the hyperactive AID*Δ mutation for random base editing[34]. The insert fragment was obtained by two sequential PCR reactions to amplify AID*Δ and additionally add a part of the SGGS-SGGS-XTEN-SG-SGGS-SGGS-linker to match the vector after restriction digest. PCR product and vector were both digested with NotI-HF (NEB) and XmaI (NEB) according to the manufacturer's protocol. Desired DNA fragments were then extracted using the QIAquick Gel Extraction Kit (QIAGEN). Ligation was then done using T4 DNA Ligase (Thermo Fisher Scientific) according to the manufacturer's protocol.

Additionally, the mutation H840A was introduced to the already pre-existing mutation D10A to eliminate the endonuclease activity of Cas9[59]. The insert, from the dCas9 plasmid (Addgene #68416) and vector fragments were obtained by digesting both plasmids with 1 μL NheI-HF (NEB) and BstEII-HF (NEB). Restriction digest reactions were loaded onto a 1% agarose gel. Desired DNA fragments were then extracted using QIAquick Gel Extraction Kit (QIAGEN) and ligated using T4 DNA Ligase (Thermo Fisher Scientific) according to the manufacturer's protocol.

**sgRNA plasmids.** sgRNAs for base editing screens of *Hprt*, *Msh2*, and *Spen* were designed so that they had a spacing of approximately 50 bp to a maximum of 100 bp between each of the PAM sites, wherever possible, and covered the whole coding region of the gene of interest. To design the sgRNAs, CRISPOR was used (Concordet, 2018). As input sequence, exonic regions with adjacent intronic regions were separately loaded into the tool and run with the reference *Mus musculus* C57BL/6 J (mm10) and the PAM 20bp-NGG for *Streptococcus pyogenes* (Sp) Cas9. sgRNAs were then selected by their efficiency and minimal off-targets. The forward (5′-CACCGNNNNNNNNNNNNNNNNNNNN-3′) and reverse (5′-AAACNNNNNNNNNNNNNNNNNNNNC-3′; N = spacer sequence without PAM sequence) oligos were annealed and then ligated with the BsmBI-digested and linearized lentiGuide-Puro vector (Addgene #52963). A list of gRNA sequences is provided in Supplementary Data 1.

**Transfection of ESCs with base editor constructs**
24 h before transfection, 100'000 cells were seeded onto a well of a 6 well-plate, and were transfected with 2 μg sgRNA plasmid pool and 500 ng of AIDhyp-dCas9-2xUGI using Lipofectamine 3000 (Invitrogen) according to the manufacturer's protocol. For cells cultured in NDiff+2iLIF (haploid ESCs), the medium was switched to ESC+2iLIF, as NDiff+2iLIF resulted in low transfection efficiency with Lipofectamine3000 (data not shown). The RNA-lipid complex was then added

dropwise to the cells. Cells were then incubated at 37 °C and 5% CO2. After 5 h, the medium was switched back to NDiff+2iLIF, where applicable. The next day, medium was changed, and cells were selected with puromycin (1 μg/mL for diploid, 0.5 μg/mL for haploid ESCs) for 48 h or 24 h, respectively. For the 48 h selection, fresh puromycin medium was added after 24 h.

## Library preparation

**Long-range PCR & NGS.** To analyze mutations obtained by the random base editing screen, mutant cells were used for targeted next-generation sequencing (NGS). Briefly, RNA of unselected or selected cell pools was extracted, and complementary DNA (cDNA) was synthesized using PrimeScript RT Master Mix (Takara Bio). cDNA was then used to amplify the coding region of the gene of interest. PCR primers were designed so that the maximum product length was around 2000 bp. PCR products overlapped by ~100 bp to avoid a gap in sequencing coverage due to the tagmentation of the PCR products. To generate sequencing libraries from the long-range PCR products, Nextera XT DNA Library Prep Kit (Illumina) was used according to the manufacturer's protocol. *Hprt* (single-end, 300 bp) and *Msh2* (paired-end, 250 bp) screen libraries were sequenced on a MiSeq System (Illumina), and *Spen* (paired-end, 150 bp) screen libraries were sequenced on a NovaSeq System (Illumina). A list of PCR primers is provided in Supplementary Data 2.

**RNA-seq.** RNA was extracted using the RNeasy Mini Kit (QIAGEN). RNA extractions were then used for library preparation with Illumina Stranded mRNA Prep, Ligation (Illumina). Libraries were sequenced as paired-end reads (150 bp) on an NovaSeq X Plus system (Illumina).

## NGS analysis

**Mutagenesis screens.** Raw reads were hard trimmed using fastp (version 0.20.1) (parameters: --cut_front –cut_front_window_size 4 –cut_front_mean_quality 30 --cut_tail --cut_tail_window_size 4 --cut_tail_mean_quality 30 --average_qual 30 --trim_poly_x --poly_x_min_len 10 --length_required 18). Sequencing reads were aligned to the cDNA sequence of the gene of interest (genome reference *Mus musculus* GRCm39). For the alignment, Bowtie2 (version 2.4.2) was used. Unmapped reads or reads containing insertions or deletions (InDels) were removed from the BAM file, since only point mutations will be considered in this analysis. For that, samtools (version 1.16.1; using htslib 1.16) view was used with the option '-F 0×4', which will remove all reads with flag 0×4 (UNMAP), i.e., unmapped reads. Samtools view and awk were used to remove any reads containing InDels (samtools view -h *input*.bam | awk '$1 ~"@" || $6 !~ "I | D"' | samtools view -b - > *output*.bam). Sequencing coverage per position was then extracted using samtools depth with the option –aa. Base changes were then analyzed and quantified using a custom python script, which (1) defines all base changes per read, (2) defines synonymous and/or non-synonymous amino acid substitutions per read, and (3) quantifies identified amino acid substitution by calculating their frequency by calculating the number of mutations relative to the given mean sequencing coverage of the amino acid (i.e. base triplet). Reads with multiple amino acid substitutions were excluded from the analysis after step (2), and sequencing coverages were adjusted accordingly.

For screen hit selection (1), substitution frequency in the selected populations and (2) substitution enrichment (frequency$_{selected}$/frequency$_{unselected}$) were ranked. (1) Substitutions with a higher frequency than the highest-ranked substitution of the unselected cell pool were considered. (2) To avoid division by zero, all zeros were replaced by 0.01, corresponding to the expected sequencing error rate. The top 15 enriched mutations were then compared to the selected substitutions ranked by frequency. (1) (2) Overlapping amino acid substitutions were considered as screen hits. Screen hit calling was done for each cell pool individually and summarized.

**RNA-Seq.** Raw data was first quality trimmed using fastp (version 0.20.1) (parameters: --cut_front –cut_front_window_size 4 –cut_front_mean_quality 30 --cut_tail --cut_tail_window_size 4 --cut_tail_mean_quality 30 --average_qual 30 --trim_poly_x --poly_x_min_len 10 --length_required 18). Reads were then aligned using hisat2 (version 2.2.1) (parameters: --dta –rna -strandness RF --very-sensitive). Reads were aligned to the genome reference *Mus* musculus GRCm39 (NCBI RefSeq assembly: GCF_000001635.27). Read duplicates were marked and removed by Picard MarkDuplicates (version 3.1.1) (parameters: --CREATE_INDEX true --VALIDATION_STRINGENCY SILENT --REMOVE_DUPLICATES true --ASSUME_SORTED true).

**Allele-specific analysis.** Working with a hybrid TX/Cast cell line (*Mus musculus* x *Mus musculus castaneus*) allowed the assignment of the two X chromosomes to their corresponding genetic background. Data was aligned to the genome reference *Mus musculus* GRCm39 (NCBI RefSeq assembly: GCF_000001635.27) and therefore contained SNP information at sites that were from the *Mus musculus castaneus* allele. SNP background was verified by analyzing allele frequencies of the TX XGFP/Cast with and without doxycycline-induced *Xist* expression.

Reads aligned to the X chromosome (RefSeq: NC000086.8) were extracted from the aligned BAM files using samtools view (version 1.16.1; using htslib 1.16). SNP calling was then done by using GATK (4.1.4.1) HaplotypeCaller. Resulting variant call format (VCF) files were then filtered for SNPs that had a minimal coverage of 10 reads by using bcftools filter (version 1.13; using htslib 1.13).

Bcftools annotate (version 1.13; using htslib 1.13) was then used to assign chromosome ID ('CHROM'), gene start ('FROM'), gene end ('TO'), and gene name ('ID') to the VCF files using a BED file obtained from Table Browser (UCSC Genome Browser, https://genome.ucsc.edu/cgi-bin/hgTables, 20.01.2025), where BED file was generated for all genes in the group ' Genes and Gene Predictions' of the table 'RefSeq All'. Note that non-curated and predicted genes were removed. Finally, SNPs were extracted from the VCF files using bcftools filter (version 1.13; using htslib 1.13). The TX and Cast allele frequencies were then obtained from the 'AD' values for each SNP (AD = "Allelic depths for the ref and alt alleles in the order listed"). Allelic frequency tables were then filtered so that they contained SNPs for which data is available in all samples compared.

**TPM calculations.** To normalize RNA-seq data to transcript per million (TPM) values, stringtie (v.3.0.0) was used. Subsequently, z-scores of each gene were calculated ($z = \frac{sampleTPM - meanTPM}{stdevTPM}$). Z-score calculations were done for each replicate independently.

## Mass spectometry

**Protein extraction.** Cell pellets were lysed in 800 μL 4 % SDS, 50 mM Tris/HCl, pH 8.2, subjected to two minutes bead beating with TissueLyzer II (Qiagen), boiled for 5 min at 95 °C, and treated with High Intensity Focused Ultrasound (HIFU) twice for one minute at an ultrasonic amplitude of 90%. The supernatant was recovered after centrifugation for 5 min at 20,000 x g. The protein concentration was then determined using the Lunatic UV/Vis polychromatic spectrophotometer (Unchained Labs).

**Protein digestion.** 500 μg of protein were reduced with 5 mM TCEP (tris(2-carboxyethyl)phosphine) and alkylated with 15 mM chloroacetamide at 30 °C for 30 min in the dark. Samples were processed using the single-pot solid-phase enhanced sample preparation (SP3). The SP3 protein purification, digestion, and peptide clean-up were performed using Carboxylate-Modified Magnetic Particles (GE Life Sciences)[60]. Beads were conditioned following the manufacturer's instructions, consisting of three washes with water at a concentration of 1 μg/μL. Samples were diluted with 100% ethanol to a final concentration of 60% ethanol. Protein capturing (30 min), washing (3×80%

Ethanol), and trypsin digestion were carried out in Eppendorf tubes on a Thermoshaker. Proteins were digested overnight at 37 °C with a trypsin:protein ratio of 1:50 in 50 mM Triethylammoniumbicarbonat (TEAB) followed by peptide elution from the magnetic beads using MilliQ water (2×200 uL). The digest solution and water elutions were combined and fully dried.

**Phosphopeptide enrichment and Evotip loading.** The phosphopeptide enrichment was performed using a KingFisher Flex System (Thermo Fisher Scientific) and Ti-IMAC HP MagBeads (ReSyn Biosciences). Beads were conditioned following the manufacturer's instructions, consisting of 3 washes with 400 μL of binding buffer (0.1 M glycolic acid, 80% acetonitrile, 5% TFA). Each fraction was dissolved in 150 μL binding buffer, and approximately 2% of each sample was taken for the whole proteome analysis. Beads, wash solutions, and samples were loaded into 96-well plates and transferred to the KingFisher Flex System. Phosphopeptide enrichment was carried out using the following steps: binding of the phosphopeptides to the beads (30 min), washing the beads in wash 1, 2 and 3 (wash buffer 1: 0.1 M glycolic acid, 80% acetonitrile, 5% TFA, wash buffer 2: 80% acetonitrile, 1% TFA, wash buffer 3: 10% acetonitrile, 0.2% TFA, 3 min each) and eluting peptides from the beads (80 μl 1% NH4OH in water, 10 min). To each elution 10 μl of 20% formic acid was added. Phospho-enriched as well as a small amount of the input fraction for full proteome analysis corresponding to approximately 200 ng were loaded onto Evotips, according to the manufacturer's instructions.

**Liquid chromatography-mass spectrometry analysis.** MS analyses were performed on a timsTOF Pro (Bruker) coupled to an Evosep One (EvoSep Biosystems). Samples were separated with the extended Evosep method "15 samples/day" keeping the analytical column (PepSep C18 15 cm×150 μm, 1.5 μm) at 50 °C. For the dual timsTOF, MS spectra were scanned from m/z 100 to m/z 1700 in ddaPASEF mode (data dependent acquisition Parallel Accumulation Serial Fragmentation). Inversed ion mobilities (1/K0) ranging from 0.60 Vs/cm$^2$ to 1.60 Vs/cm$^2$ were analyzed with ion accumulation and ramp time of 100 ms, respectively. 1 survey TIMS-MS scan was followed by 10 PASEF ramps for MS/MS acquisition, resulting in a 1.17 seconds cycle time. Singly charged ions were excluded using the polygon filter mask, and isolation windows for MS/MS were set to m/z 2.0 for precursor ions below m/z 700, and m/z 3.0 for ions above and fragmented with default fragmentation energy settings. The mass spectrometry proteomics data were handled using the local laboratory information management system (LIMS)[61].

**Protein and phosphopeptide identification.** The acquired shotgun MS data were processed for identification and quantification using the FragPipe computational pipeline (version 22.0, (https://fragpipe.nesvilab.org/). Spectra were searched against a Uniprot *Mus musculus* reference proteome (UniProt ID UP000000589, reviewed canonical version from 2025-07-03), concatenated with reversed sequences and common protein contaminants, using MSFragger 4.1 and Percolator. Carbamidomethylation of cysteine was specified as a fixed modification, while methionine oxidation and phospho serine, threonine, and tyrosine (for enriched samples) were set as variable modifications. Enzyme specificity was set to trypsin/P allowing a minimal peptide length of 6 amino acids and a maximum of two missed cleavages. Precursor ion mass tolerance was set to ±20 ppm, and fragment ion mass tolerance was set to 20 ppm. Default Philosopher (version 5.1.1) ion, peptide, and protein FDR values (0.01%, 1%, 1%) were used, and phosphosite re-scoring was performed using the integrated ptmProphet node with default settings. MS1 quantification was performed by IonQuant (version 1.10.27) with MaxLFQ activated. For protein-level analysis, proteins were typically required to be identified with a minimum of two unique peptides. The list of identified phosphopeptides is listed in Supplementary Data 3.

**Engineering the W522C substitution of the endogenous *Spen***

W522C substitution was re-constituted in female TX XGFP/Cast ESCs (*Mus musculus* x *Mus musculus castaneus*) using CRISPR/Cas9 and donor-guided homology-directed repair (HDR). Single-stranded donor templates (+and − strand) and gRNAs were designed using Alt-R CRISPR HDR Design Tool (IDT, https://eu.idtdna.com/pages/tools/alt-r-crispr-hdr-design-tool, 09.12.2024). gRNA sequences and donor sequences are provided in Supplementary Data 1.

gRNA was complexed by combining trRNA and crRNA in a 1:1 ratio and incubating the mixture at 95 °C for 5 min with subsequent cooling to room temperature on the bench top. The RNP complex was made by combining 120 pmol Cas9 protein with 3 μL gRNA complex (50 μM) and incubating at room temperature for 10-20 min.

Cell suspension containing 10$^6$ ESCs, 8 μL RNP, 2 μL of either + or − donor, 2 μL electroporation enhancer (IDT, manufacturer), and Opti-MEM (Gibco) in a total volume of 100 μL were electroporated using a NEPA21 Type II electroporator (Nepa Gene Co.) (settings: Poring pulse: 135 V, 5 ms pulse, 50 ms pulse interval, 2 pulses, 10% decay rate, + polarity; Transfer pulse: 20 V, 50 ms pulse, 50 ms pulse interval, 5 pulses, 40% decay rate, +/− polarity). Electroporated cells were plated in gelatine-coated plates containing 1.7 μL HDR enhancer (IDT manufacturer) per 1 mL medium and incubated at 37 °C and 5% CO$_2$. After 24 h, the medium was changed to normal culture medium. Cells were again incubated at 37 °C and 5% CO$_2$ until colonies were large enough for manual subcloning. Subclones were seeded into wells of 96-well plates and genotyped.

**Generation of ESCs with C-terminally tagged endogenous SPEN**
To generate cells containing endogenously tagged SPEN, ESCs were transfected using Lipofectamine3000 (Invitrogen) with a Cas9-PuroR (pX459) vector containing a SPEN-targeting sgRNA (5′-ATTGT-CATTGCCTCGGTGTG-3′) and plasmid Spen.GFP_HA.GFP.PuroR.HA containing a donor template with homology arms of 600 bp containing an RFP gene (both plasmids were kindly provided by Prof. Dr. Joost Gribnau; the original *eGFP* was replaced by an *RFP* using the NheI and AscI restriction sites flanking the *eGFP* gene) as previously done by Robert-Finestra et al. (2021)[52]. The donor template additionally contained a loxP-flanked puromycin-resistance gene, allowing for stable introduction of the donor template by selecting the cells for three passages with 0.5 μg/mL puromycin. After selection, RFP-positive cells were sorted as single cells by FACS. Genotyping was done using two primer pairs to separately amplify the *RFP* gene (forward primer 5′-AGTTCTCCGAGAGTCACCTC-3′ combined with 5′-AGGACAGCTTC AAGTAGTCG-3′ (within the *RFP* gene) or with 5′-TTGTAGCCGCGTTCT AACGA-3′ (outside the *RFP* gene; used for sequencing) and a primer pair to amplify the wild-type sequence (forward primer: 5′-TTGGTCA TTGTCCCATCCCT-3′; reverse primer: 5′-TTAGTGACTCGCGCAGTGAA-3′) of the same region. The RFP sequence was verified by Sanger sequencing in cells with no additional wild-type PCR products.

**Generation of SPEN-/- ESCs**
Homozygous *Spen*-knockout (SPEN-/-) ESC lines were generated using CRISPR/Cas9 technology. Cells were electroporated with two sgRNAs that target the 5′ ((5′-TCGGACAAGACATTACGATC-3′) and 3′ (5′-TGTGGACGACGCAAGTGCAC-3′) region of the *Spen* ORF complexed with Cas9 to form ribonucleoprotein (RNP) complexes. Cells were electroporated using a NEPA21 Type II electroporator (Nepa Gene Co.) (settings: Poring pulse: 135 V, 5 ms pulse, 50 ms pulse interval, 2 pulses, 10% decay rate, + polarity; Transfer pulse: 20 V, 50 ms pulse, 50 ms pulse interval, 5 pulses, 40% decay rate, +/− polarity). Cells were then sub-cloned and genotyped. Primer 3-F (5′-GAAAGAGGCGAGG CGTAAAG-3′) was combined with 3-R (5′-CGGGGACTGGTTTCTAG ACT-3′) or 12-R (5′-CTCGGTCATTTTCCTGCTCG-3′) to amplify and detect wild-type or knockout sequences, respectively.

## RNA FISH

Cells were centrifuged onto microscope slides using a Cytospin 4 (Thermo Fisher Scientific) funnel. Slides were then placed into (1) PBS (Gibco), (2) CSK (100 mM sodium chloride, 300 mM D(+)-Saccharose sucrose, 3 mM magnesium chloride, 10 mM PIPES (pH 6.8)) for 30 s, (3) CSK + 0.5% Triton X-100 (ITW Reagents Division), (4) CSK for 30 s, (5) 4% paraformaldehyde (Santa Cruz Biotechnology) for 10 min, and (6) in 70% EtOH. After, slides were dehydrated in an EtOH series of 70%, 80%, 95%, and 100% for 2 minutes each. Slides were then air-dried. The Xist-specific RNA FISH probe was then diluted with Hybrisol (MP Biomedicals) and added to the cell sample, which was incubated at 37 °C overnight. Slides were then washed in (1) 4X SSC (20X SSC: 3 M sodium chloride, 0.3 M tri-sodium citrate dihydrate) and formamide (50% v/v) for 15 min at 39 °C, (2) three times 2X SSC for 5 min at 39 °C, and (3) 1X SSC for 10 min at room temperature. Slides were then mounted by adding a drop of VECTASHIELD containing DAPI (Vector Laboratories, Inc.).

## IF-FISH

Cells were seeded onto wells containing coverslips, which were coated with Geltrex (Thermo Fisher Scientific). After 4 d of Dox treatment, slides were washed with PBS (Gibco) and fixed with 4% paraformaldehyde (Santa Cruz Biotechnology) for 10 min. Cells were washed twice with PBS (Gibco) and subsequently permeabilized for 10 min on ice in PBS containing 0.5% Triton-X (ITW Reagents Division) and 0.1% sodium citrate. Cells were washed twice for 30 s with ice-cold PBS containing 0.1% TWEEN 20 (Sigma-Aldrich) (PBS-T). Then, cells were blocked by incubating them in PBS-T containing 2.5% BSA for 45 min. Cells were then incubated for 1 h with primary antibody solution (EZH2 (Cell Signaling Technology): 1:200; H3K27me3 (Active Motif): 1:200; in blocking solution). Cells were washed three times with PBS (Gibco), and the secondary antibody solution (anti-rabbit Alexa488 (Jackson ImmunoResearch): 1:1000; in blocking solution) was added and again incubated at room temperature for 1 h. Cells were rinsed twice with PBS-T and once with PBS. To further perform RNA FISH on the samples, slides were post-fixed with 4% PFA for 10 min and washed once with PBS and once with 2X SSC. After the washing steps, the slides were air-dried before the RNA FISH protocol was continued with the overnight hybridization.

## Microscopy

**Image acquisition.** All microscopy images were captured with a Hamamatsu Camera (C11440-22C) mounted on a Zeiss Observer Z1 inverted widefield microscope equipped with a Plan-Apochromat 40x/ 1.4 Oil DIC (UV) VIS-IR M27 objective. Images were acquired as z-stacks at a color depth of 16 bits per pixel and a spatial resolution of 2048 × 2048 pixels.

Cy3 (used for RNA FISH on Xist RNA) was imaged at max. excitation wavelength of 548 nm and with a max. emission wavelength of 561 nm. DAPI was imaged at max. excitation wavelength of 353 nm and with a max. emission wavelength of 465 nm. eGFP and RFP (live-cell imaging) were captured at max. excitation/emission wavelengths of 488/509 nm and 590/612 nm, respectively.

**Image processing.** All microscopy image processing was performed using ImageJ (1.53t). For representative images, display contrast was adjusted by linearly rescaling the full image intensity range through global minimum and maximum thresholding. No pixel-level, regional, or nonlinear modifications were performed. Z-stacks were collapsed into two-dimensional images using maximum intensity projection across all optical sections.

Cy3 area calculations to quantify Xist RNA areas per nucleus were done by changing both images containing DAPI-stained nuclei and Cy3-stained Xist RNA clusters to 8-bit images. Thresholds were adjusted individually; for nuclei, automatic thresholding was used, while for

Xist clusters lower threshold was set to 0.15% for all conditions analyzed. Areas were then calculated using the Fiji plugin for particle analysis.

## Cut&Run

CUT&RUN was performed using the CUT&RUN Assay Kit (Cell Signaling Technology) and an antibody against H3K27ac (Cell Signaling Technology, 1:50). DNA was purified from enriched chromatin samples using Spin Columns (Cell Signaling Technology). For library prep, NEBNext Ultra II DNA Library Prep Kit for Illumina (NEB) was used. Libraries were sequenced on an Illumina MiSeq instrument as 150 bp paired-end reads. Fastq files were aligned to the reference genome (GRCm38.6) using Bowtie2 (version 2.4.2).

## Native ChIP

ChIP was performed using an antibody against H3K27me3 (Tri-Methyl-Histone H3 (Lys27) (C36B11) Rabbit Monoclonal Antibody; Cell Signaling, 3:500). Between 1 and 4 million ESCs per immunoprecipitation (IP) were pelleted and resuspended in 90 μL of Lysis Buffer (LB) (50 mM TrisHCl pH 7.5, 150 mM NaCl, 0.1% sodium deoxycholate, 1% Triton, 5 mM CaCl2, 1X PIC). Chromatin was digested using 60 μL of LB containing 0.3 μL Micrococcal nuclease (Cell signaling) at 37 °C for 15 min. Reactions were stopped with 15 μL 0.5 M EDTA, and Stop Buffer (SB) (50 mM TrisHCl pH 7.5, 150 mM NaCl, 0.1% sodium deoxycholate, 1% Triton, 30 mM EGTA, 30 mM EDTA, 1XPIC). After adjusting the volume to 1 ml with LB/ SB (1:1), 30 μL were used as input. 300 μL of chromatin was adjusted to 1 ml with LB/ SB (1:1) and incubated over night with corresponding antibody at 4 °C. 15 μL Dynabeads protein G (Thermo Fisher Scientific) blocked with blocking buffer (PBS 1X, 0.5% Tween, 0.5% BSA) were added and incubated 2 h at 4 °C. After washing, DNA was eluted from beads using ProtK Buffer (20 mM HEPEs, 1 mM EDTA, 0.5%SDS), and digested for 2 h at 56 °C. Input samples were also resuspended in ProtK buffer and digested. Samples were then purified using the Zymo ChIP DNA Clean Concentrator kit (Zymo Research). Purified DNA was used for library preparation using the NEBNext ultra II DNA Library Prep Kit from NEB (NEB). Libraries were sequenced on an Illumina NovaSeq X machine using 150 bp paired-end sequencing.

## Statistics & Reproducibility

No statistical method was used to predetermine sample size. No data were excluded from the analyses done in this manuscript.

Unless otherwise stated, 'n' values indicate independent biological replicates. The experiments were not randomized. The authors were not blinded to allocation during experiments and outcome assessment, except for image-based quantifications, in which cells were selected based on DAPI staining prior to blinded fluorescence signal quantification.

Statistical analyses were performed as described in the figure legends. Multiple-testing correction was applied where indicated. Data analyses and plotting were performed using RStudio (v.2024.09.1 + 394) running R v.3.5.0, PyCharm (v.2021.3.1) running Python (v3.10), and GraphPad Prism 10 (v.10.1.0).

## Reporting summary

Further information on research design is available in the Nature Portfolio Reporting Summary linked to this article.

# Data availability

All raw sequencing data from mutagenesis screens on *Hprt*, *Msh2*, and *Spen* generated in this study have been deposited in the Sequence Read Archive of the National Center for Biotechnology Information (NCBI) database with BioProject number PRJNA1265691.

All raw sequencing data obtained in this study from RNA-seq, ChIP, and Cut&Run for the functional analysis of the mutation of SPEN residue W522 have been deposited to the Sequence Read Archive of

the National Center for Biotechnology Information (NCBI) database with BioProject no. PRJNA1380311.

Mass spectrometry data obtained in this study have been deposited to the Proteomics Identifications Database (PRIDE) with the project accession number PXD073932.

Source data are provided with this paper. A list of reagents used in this study is provided in Supplementary Table 1. Source data are provided with this paper.

## Code availability

Python script, and intermediate data for computational analyses of the CRISPR/Cas9-guided base editing screens on *Hprt*, *Msh2*, and *Spen* are available on GitHub at https://github.com/CorinneKa (https://doi.org/10.5281/zenodo.18475593). Python script, genomic annotation, and intermediate data for computational analyses of Cut&Run (https://doi.org/10.5281/zenodo.18393627) and ChIP (https://doi.org/10.5281/zenodo.18393601) experiments are available on GitHub at https://github.com/WutzLab.

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

## Acknowledgements

Genomics and proteomics were performed with the support of the Functional Genomics Center Zurich (FGCZ) of the University of Zurich and ETH Zurich. The authors gratefully acknowledge the FGCZ, and in particular Dr. A. Dittmann, for expert support with the proteomics experiments performed in this study. We are grateful to Prof. Dr. J. Gribnau (Erasmus University) and Dr. T. Robert-Finestra (Josep Carreras Leukemia Research Institute) for providing the Cas9-PuroR (pX459) and Spen.GFP_HA.GFP.PuroR.HA plasmids used in this study.

This work was supported by the Swiss National Science Foundation (SNSF; grant 31003A_152814/1).

## Author contributions

C.K. and A.W. planned the study. C.K. performed most experiments. C.K. analyzed and interpreted most NGS data. S.S. performed native ChIP and Cut&Run experiments and their analysis/interpretation. S.S. and C.K. generated RFP-tagged cell lines used in this study. C.D. generated and provided cell lines containing homozygous deletions of the RRM regions of SPEN (SPEN-/- 1 and 2). C.K., S.S. and A.W. wrote and revised the manuscript. All authors approved the final manuscript.

## Competing interests

The authors declare no competing interests.
