## [Transparent Peer Review file · Nature Communications]

Comprehensive CRISPR/Cas9-based mutagenesis identifies single-amino acid substitutions that abrogate SPEN function in X inactivation

Corresponding Author: Professor Anton Wutz

Version 0:

Reviewer comments:

Reviewer #1

(Remarks to the Author)

In this paper, the authors developed base editor-guided point mutagenesis screening of genes on X chromosome and autosome using mouse diploid and haploid embryonic stem cells (ESCs), respectively. Using this technology, the authors finally identified an important role for SPEN W522 residue in the X chromosome inactivation.

In the first part of this paper, a new base editing screening technique is explored. Today, screening using base editor is widely used method; however, the method of analyzing the function by comprehensively performing base substitutions by placing sgRNAs every 50 bp is a novel application. On the other hand, using mouse haploid ESCs to analyze the function of genes on autosomes may not necessarily be an outstanding method to develop, since there is already a report of base editing screening using near haploid cell line HAP1 in humans (PMID: 33606977).

The key findings of this paper are in the latter part of the paper. The functional analysis of SPEN is highly novel and timely topic. The authors have identified important amino acid mutations involved in SPEN function by base editing screening. The authors used haploid ESCs with a Dox inducible Xist allele. A method in which ESCs cannot survive when XCI is induced is very sophisticated. Using this system, the authors have found several important amino acid substitution mutations (E448, T450, A489, L503 of RRM3 and W522 and T536 of RRM4) by base editing screening. After the detailed analysis, the authors clarified that the substitution of the SPEN RRM4-residue W522 abrogates X-linked gene repression by Xist RNA and impairs H3K27-deacetylation, but does not affect Polycomb recruitment and H3K27me3 deposition.

Most studies were properly executed, and data are convincing; however, the reviewer have two questions, which are described below.

1) Even though the authors found many SPEN mutations involved in XCI (Fig. 3C), only the W522 was examined in detail. This somewhat limits the results of this paper. Why did the authors not proceed with a detailed analysis of other amino acid residues?

2) Is there any possibility that the method of designing sgRNAs every 50 bp (up to 100 bp) miss any important mutations?

(Remarks on code availability)

Reviewer #2

(Remarks to the Author)

In this manuscript, the Wutz group developed and validated a new dCas9-based approach for saturating mutagenesis of coding regions in mouse embryonic stem cells. They cleverly link a deaminase version (based on AID) and two UIG domains to dCas9 to enable their sequence-specific targeting while reducing base excision repair. They validate this tool using three different approaches: mutagenesis of HPRT in male ESCs (an X-linked gene that in male cells only exists in one copy), and of MSH2 and SPEN, autosomal genes, in haploid mouse ESCs. The use of haploid ESCs allows the authors to

overcome the detection of only dominant negative mutations. It is interesting to me that they can saturate mutagenesis in the coding region with a limited number of gRNAs targeting the machinery to about 50bp intervals of interest. The SPEN mutagenesis is directly testing for mutations that interfere with X chromosome inactivation. Beyond developing and validating the new dCas9 tool, the authors focus their work on the dissection of the silencing function of SPEN on X-chromosome inactivation. They identify one critical mutation that interferes with silencing, and they use this mutation to then show that SPEN's silencing function and the Polycomb pathway are independent pathways, which has been an unresolved question. The work seems straight forward and is well written. Surprisingly, the authors only follow up on one mutation of SPEN, for that the authors do not show why SPEN does not work as the silencing protein anymore, and the SPEN follow-up work generally shows some weaknesses as described in the comments below.

Specific Comments:

The W522 mutation is recapitulated with only one independently generated mESC line. Since this mutation is also used for all functional studies in this paper, it would be important to see the data repeated with an independent cell line.

Similarly, the mutation is only tested in the context of tetO Xist and not normal female mouse embryonic stem cells during differentiation, which would be interesting to see.

The authors also speculate that the mutation blocks RNA binding by SPEN but unfortunately do not characterize their mutation that way.

In Figure 4C, there is an Xist signal in the no dox control. Is this Tsix or leaky Xist expression? It also seems that 4 days of dox induction of Xist expression is very long, as Xist is filling nearly the entire nucleus? Moreover, the images in Figure 4C suggest that the Xist signal is more widely distributed in the SPEN mutant line – is that the case? It seems the authors argue there is less Xist, but the image then is misleading.

The gene silencing data could be better presented with statistics and data quantification (in Figure 4D and 4E). It is for instance unclear if there is any silencing in the SPEN mutant line. The data are somewhat undetailed.

The chromatin data in Figure 5 also lack any statistics and those for the CpG islands are particularly unclear. For the immunostaining of H3K27me3 and EZH2 not only the proportion of cells with enrichment should be considered but also the intensity of the enrichment.

Minor Points:

At the bottom of page 23 and a few other places – are the authors referring to Table 3 rather than Table 4?

Is there a table for the phosphoproteomics data that lead to the identification of S1284 phosphorylation?

(Remarks on code availability)

Reviewer #4

(Remarks to the Author)

In this study the authors develop a CRISPR/Cas9-AID* saturation mutagenesis strategy for analysing the function of individual genes of interest in mammalian cell culture models. They validate their system performing mutagenesis and 6TG selection for the X-linked Hprt gene in hemizygous XY ES cells and the MSH2 gene in haploid ES cells. They show that substitutions can be recovered for the majority of individual residues in both proteins, and that under selection, predominating mutations map to known functional domains and/or overlap with disease mutations seen in human disease attributable to the cognate genes. They go on to perform mutagenesis on the gene encoding the protein SPEN, a key factor for chromosome silencing in X chromosome inactivation. For this they make use of an established haploid XY ES cell line in which X inactivation can be induced using a doxycycline inducible Xist allele. Xist induction in these cells is ~ 90% cell lethal but SPEN loss of function rescues viability. Again, mutations are found across the length of this large (400kda protein), and following selection (Xist induction) several interesting substitutions are identified in SPEN functional domains, including the very extensive IDR regions of the protein. They go on to show that one of the IDR mutations, S1284 is a site with high probability of being phosphorylated, and indeed confirm phosphorylation of this site specifically in cells with Xist RNA induction using phosphoproteomics. Finally, the authors investigate an additional function-linked SPEN mutation, W522, which occurs in the RNA binding RRM3-4 region of the protein in greater detail. They reproduce the W522 mutation in an XX mESC line to enable detailed analysis of X inactivation phenotypes. They show that W522 mutation results in a strong effect on SPEN function in gene silencing and in promoting histone deacetylation (H3K27Ac). They also observe decreased levels of Xist RNA in the SPEN mutant line, potentially linked to an RNA stability defect. They see no effect on the ability of Xist to recruit the PRC2 polycomb complex, contrasting with prior studies that suggested SPEN loss of function reduces or abrogates PRC2 and associated H3K27me3 in X inactivation. Based on this latter observation the authors suggest that W522 represents a separation of function mutation.

Overall this is an excellent and well conducted study. The strategy for saturation mutagenesis of genes of interest in mammalian cells represents a powerful tool that will be of interest across many different fields. Similarly, detailed functional maps of HPRT, MSH2 and SPEN provide a valuable resource for understanding how these proteins contribute to fundamental cellular processes and in human disease. There are nevertheless some points that need to be addressed, most notably in relation to the W522 mutation and how these findings should be interpreted in the context of previously published studies:

1. It is not straightforward to reconcile how different studies have interpreted the effect of SPEN loss of function on Polycomb recruitment to Xi. Several different strategies have been applied to abrogate SPEN function, including RNAi knockdown, degron mediated acute depletion (with and without transgene complementation using full length or truncated constructs), in frame deletion of large domains of the protein, and engineered point mutations, notably of the SPOC domain. Added to this, different methods used to assay Polycomb on Xi range from IF for PRC2 and PRC1 subunits and/or of their cognate histone

modifications (using various antibodies that likely have different avidity/affinity for epitope), and allelic ChIP-seq analysis, in most cases in interspecific XX mESC models. Moreover, different studies have used different timepoints of Xist expression either with or without differentiation, and different modes of Xist induction (native promoter vs TetOn system). The broad consensus from these studies is that SPEN mutations result in reduced but not loss of Polycomb on Xi. Both indirect (reduced gene silencing countering Polycomb deposition) and direct (for example reduced levels/stability of Xist RNA) mechanisms could account for this (discussed also in <https://doi.org/10.1016/j.devcel.2021.04.009>). The authors of this study conclude that Polycomb recruitment is unaffected in W522 XX ES cells. However, the assays they use, scoring presence and absence of domains after IF, cannot be said to be very quantitative. The experiments lack a critical positive control, XX mESCs with a SPEN loss of function mutation that has been reported to reduce PRC2 recruitment, analysed with the same reagents and at the same induction timepoint as in this study (I actually couldn't find mention of how long Xist was induced in the experiments herein, only for the gene-silencing analysis. I assume the histone modification analysis was also with 4 days induction?).

In addition to including an appropriate control, the authors should note that a prior study using ChIP-seq reported that Polycomb modification distribution across Xi is highly aberrant in SPEN null cells (thought to be due to aberrant Xist RNA localisation), but interestingly, not in cells with point mutations that abrogate SPEN-mediated gene silencing by blocking NCoR-HDAC3 interaction with the SPOC domain (doi: 10.1126/science.abe7500). This incidentally was also the first study to document that SPEN loss of function results in reduced Xist RNA stability and low Xist RNA levels and should be referenced in this context. Given that the authors have available interspecific XX mESCs with the W522 mutation, it would seem a good idea to perform a similar ChIP-seq experiment to determine if distribution of Polycomb modifications (H3K27me3) over Xi resembles that seen in wild-type (or SPEN SPOC mutant) or SPEN null cells. A further advantage of performing this experiment is it would allow for a more quantitative assessment of Polycomb deposition, eg using spike in or normalising to Xa signal.

If as a result of these experiments it should emerge that W522 essentially phenocopies SPEN null mutations (as opposed to showing separation of function for the Polycomb phenotype) this would not in my view undermine the significance or value of this study. The availability of a range of specific point mutations affecting SPEN function will without doubt be of great value to the field as a whole.

2. The finding that S1284 phosphorylation occurs only in cells with Xist induction is curious given that SPEN likely functions at other sites as well (maybe not in ES cells?). Do the authors have a preferred hypothesis for SPEN regulation by phosphorylation that they could add to the discussion?

3. Decreased levels of Xist RNA in SPEN W522 mutant mESCs is shown only as shades of green on a heat map (as far as I could find) and I think it would be useful to have a more quantitative measure, eg qRT-PCR.

(Remarks on code availability)

Version 1:

Reviewer comments:

Reviewer #1

(Remarks to the Author)

I am satisfied with the revisions that have been made by the authors.

(Remarks on code availability)

Reviewer #2

(Remarks to the Author)

I appreciate the extensive edits and additions made by the authors and am satisfied with them. They help to better understand the impact of the W522 mutation in SPEN.

(Remarks on code availability)

Reviewer #4

(Remarks to the Author)

The authors have included new data and analysis RE SPEN mutations, and have revised conclusions somewhat, largely addressing the issues that I raised in my original review. Some points are not fully resolved but addition of some caveats and a couple of minor corrections should be sufficient to deal with these;

1. Introduction lines 34 and 35 do not capture all points from literature on SPEN effects on polycomb. As I remember the two referenced papers looked only at H3K27me3 so generalising with the term Polycomb is not factually correct. Reference 51 reported on both H3K27me3 and H2AK119ub1 in SPEN RRM and SPOC mutations, observing that both modifications are

- enriched on Xi in the RRM and SPOC mutation lines but that the RRM mutation affects distribution of both modifications on Xi, implying an effect on where Xist localises (see also below).
2. Results, line 172 states 'exhibited slight X chromosomal gene expression...' – shouldn't this be other way round, ie. slight gene repression?
 3. Line 177 mentions lower Xist levels in W522C/delta but isn't this also the case for W522C/Ins?
 4. Line 187-190. I didn't understand why the authors confined their H3K27me analysis to transcription units given gain of polycomb modifications on Xi occurs widely over CREs and intergenic regions. As noted in the rebuttal letter loss of H3K27me3 over transcription units is expected to correlate with the transcription that occur on the inactive X in SPEN mutants. Also reference 51 observed changed distribution of polycomb modifications in SPEN RRM but not SPOC mutants. As a minimum this section needs to be more carefully worded to state 'H3K27me3 over transcription units'.
 5. Line 198. The FISH analysis showing clusters are unaffected in SPEN mutations needs to be interpreted cautiously as it does not assess where Xist localises and there are caveats in the methodology where sparsely distributed molecules in mutant cells may artificially increase apparent size of clusters. The authors acknowledge this point in rebuttal but it seems to be lost in the manuscript. As a minimum a caveat to say that whilst Xist localisation appears normal in the SPEN mutants, the assays used do not assess whether distribution over the chromosome changes.
 6. Consider swapping the position of the two paragraphs in discussion as would seem more logical to first discuss the findings relating to saturation screening and then finishing with the discussion of SPEN analysis?
 7. Line 689 figure legends, missing H3K27me3 in heading.

(Remarks on code availability)

Point to point response to the reviewer comments

Response to Reviewer #1 Remarks to the Author

In this paper, the authors developed base editor-guided point mutagenesis screening of genes on X chromosome and autosome using mouse diploid and haploid embryonic stem cells (ESCs), respectively. Using this technology, the authors finally identified an important role for SPEN W522 residue in the X chromosome inactivation.

In the first part of this paper, a new base editing screening technique is explored. Today, screening using base editor is widely used method; however, the method of analyzing the function by comprehensively performing base substitutions by placing sgRNAs every 50 bp is a novel application. On the other hand, using mouse haploid ESCs to analyze the function of genes on autosomes may not necessarily be an outstanding method to develop, since there is already a report of base editing screening using near haploid cell line HAP1 in humans (PMID: 33606977).

The key findings of this paper are in the latter part of the paper. The functional analysis of SPEN is highly novel and timely topic. The authors have identified important amino acid mutations involved in SPEN function by base editing screening. The authors used haploid ESCs with a Dox inducible Xist allele. A method in which ESCs cannot survive when XCI is induced is very sophisticated. Using this system, the authors have found several important amino acid substitution mutations (E448, T450, A489, L503 of RRM3 and W522 and T536 of RRM4) by base editing screening. After the detailed analysis, the authors clarified that the substitution of the SPEN RRM4-residue W522 abrogates X-linked gene repression by Xist RNA and impairs H3K27-deacetylation, but does not affect Polycomb recruitment and H3K27me3 deposition.

Most studies were properly executed, and data are convincing; however, the reviewer have two questions, which are described below.

Response: We thank the reviewer for finding our study of interest and the comments.

1) Even though the authors found many SPEN mutations involved in XCI (Fig. 3C), only the W522 was examined in detail. This somewhat limits the results of this paper. Why did the authors not proceed with a detailed analysis of other amino acid residues?

Response: We thank the reviewer for the comment and agree that our dataset provides a resource for understanding SPEN function in X chromosome inactivation. We have focused on W522C as this mutation is predicted to be immediately relevant to the interaction between Xist and SPEN, which could allow a separation of functions of SPEN in other biological pathways that are mediated by protein interactions. Our extensive data on characterization of SPEN W522C also provides an example on how to characterize these mutations. As XCI is a complex process that involves chromatin transitions and effects on the transcription unit of X-linked genes a meaningful characterization requires to consider mechanistic details. We have decided to provide a deeper understanding on W522C at this time and feel that we would not be able to generate sufficient abundant data for characterizing a representative subset of our resource at this time. The characterization of a comprehensive set of the point mutations will however be a focus of ongoing research and has been the motivation for undertaking the screen initially. We hope the reviewer can agree to our suggestion that our characterization of W522C validates our screen and it would be difficult to obtain a robust view of the characteristics of the individual mutations from a set of straight forward set of experiments as X inactivation involves several relevant mechanisms.

2) Is there any possibility that the method of designing sgRNAs every 50 bp (up to 100 bp) miss any important mutations?

Response: We would like to respond to the reviewer in the following manner. Although, from our sequencing data it is clear that around 90% of amino acids have been mutated in our three screening experiments, we do not claim saturation. At present mutagenesis is limited to C to T transitions (G to A on the complementary strand of DNA) and, thus, some codon changes will not be possible. In addition as the reviewer correctly points out the location of gRNA binding sites within the exonic sequences influence the likelihood of inducing base changes. We have varied the gRNA spacing between 50 and 100 nucleotides and find that it is not the limiting parameter. Likely the ability to introduce base changes also depends on the efficiency of gRNA binding. Hence, a spacing of 100 nucleotides appears adequate to obtain a high coverage of the exonic sequence in mouse ESCs. We would like to point out that our screen can not investigate all possible mutations that could be constructed theoretically by replacing each position with any of 19 amino acids. Consequently, the pool of mutations is limited in the chemical diversity which can be selected from. For these reasons it is difficult to see how complete saturation of a point mutation screen could be achieved. However, single amino acid mutations that affect protein function in mammalian pathways are scarcely explored and, thus far, methodology was lacking to identify such mutations in a systematic manner. We demonstrate this by identifying 50 high confidence mutations

in SPEN that affect its function in XCI (Table 3). Extrapolating the value of point mutations from non-mammalian model systems suggests that our screening approach could overcome a current key limitation of bridging abundant genetic screening data to mechanistic function and protein biochemistry in mammalian cell biology. We believe this is where our method will contribute in future studies in a wide range of biological questions.

Response to Reviewer #2 Remarks to the Author

In this manuscript, the Wutz group developed and validated a new dCas9-based approach for saturating mutagenesis of coding regions in mouse embryonic stem cells. They cleverly link a deaminase version (based on AID) and two UIG domains to dCas9 to enable their sequence-specific targeting while reducing base excision repair. They validate this tool using three different approaches: mutagenesis of HPRT in male ESCs (an X-linked gene that in male cells only exists in one copy), and of MSH2 and SPEN, autosomal genes, in haploid mouse ESCs. The use of haploid ESCs allows the authors to overcome the detection of only dominant negative mutations. It is interesting to me that they can saturate mutagenesis in the coding region with a limited number of gRNAs targeting the machinery to about 50bp intervals of interest. The SPEN mutagenesis is directly testing for mutations that interfere with X chromosome inactivation. Beyond developing and validating the new dCas9 tool, the authors focus their work on the dissection of the silencing function of SPEN on X-chromosome inactivation. They identify one critical mutation that interferes with silencing, and they use this mutation to then show that SPEN's silencing function and the Polycomb pathway are independent pathways, which has been an unresolved question. The work seems straight forward and is well written. Surprisingly, the authors only follow up on one mutation of SPEN, for that the authors do not show why SPEN does not work as the silencing protein anymore, and the SPEN follow-up work generally shows some weaknesses as described in the comments below.

Response: We thank the reviewer for the interest in our study and the comments that have helped to further strengthen our conclusions and clarify the text of our revised manuscript.

Specific Comments:

The W522 mutation is recapitulated with only one independently generated mESC line.

Since this mutation is also used for all functional studies in this paper, it would be important to see the data repeated with an independent cell line.

Response: We thank the reviewer for suggesting the replication experiment. During our revision we have isolated a second clone that carries the W522C mutation and a frameshift mutation in a position close to W522 on the second allele. In the revised version new experiments are included that analyze gene silencing and chromatin changes after Xist induction in the two cell lines. We have also included two newly derived cell lines which carry homozygous deletions of the entire RRM domains of SPEN, which we and others have shown to lead to a complete loss of gene repression by Xist. We use these cell lines to further characterize the relative effect of W522C. These experiments are now included in new Fig. 4 and 5 and contain a substantial amount of new data that we hope will address the concern.

Similarly, the mutation is only tested in the **context of tetO Xist and not normal female mouse embryonic stem cells** during differentiation, which would be interesting to see.

Response: We thank the reviewer for the suggestion of the experiment, which we have performed during our revision by differentiating our SPEN W522C mutant cell lines in EpiLC medium, which induced upregulation of Xist and initiation of X inactivation. The new data are shown in Fig. 4C-F. Briefly, we observe that Xist is induced in cells carrying the W522C mutation suggesting that a counting mechanism is independent of W522. However, repression of X-linked genes cannot be observed in contrast to wild type cells suggesting that the SPEN W522C mutation is not able to support gene repression by Xist. We have to point out that our analysis is on overall X-linked gene expression as random choice prevent a deterministic allelic analysis as either the *Mus castaneus* or TX X chromosome can be chosen for inactivation. Therefore, X linked gene expression is in the presence of a fully active X chromosome and, thus, silencing is detected with lower sensitivity than in our allelic analysis, where the TX chromosome is silenced by doxycycline induced Xist and the *castaneus* polymorphisms serve as a reference for the active X. Irrespective of this minor technical limitation our new data confirm our earlier conclusions and are consistent with the view that SPEN W522C impairs gene repression by Xist.

The authors also speculate that the mutation blocks RNA binding by SPEN but unfortunately do not characterize their mutation that way.

Response: We agree with the reviewer that experimental testing of the effect of W522C on SPEN interaction with Xist is important. To address interaction of SPEN W522C in cells we have engineered an RFP tag into the C-terminal position of the endogenous gene locus and provide new data on localization and accumulation of SPEN wild type and W522C mutant proteins in living cells in new Fig. 5. Our data show that the mutant protein can localize to Xist clusters, but the formation of SPEN clusters is drastically reduced compared to wild type SPEN. We interpret this observation in the following way. W522 is located in a critical position within RRM4 and likely affects the affinity of RRM4 for Xist. As a consequence the interaction of SPEN W522C is weakened and results in a lower amount of protein bound to the A repeat. Alternatively, the interaction might be less stable resulting a more rapid unbinding and dissociation of SPEN W522C from Xist. To discriminate between the two possibilities measurement of the on and off rates would be necessary which we deem beyond the scope of our present study as suitable experimental systems for quantitative measuring the SPEN - repeat A RNA binding will be needed.

In Figure 4C, there is an Xist signal in the no dox control. Is **this Tsix or leaky Xist expression?**

Response: In our Xist RNA FISH experiments we use random priming labeled fluorescent probes that do not allow strand specific assignment. From the literature we would expect biallelic Tsix expression and hence attribute at least part of the dot like signal to Tsix. However, we cannot rule out that low transcription of Xist also does take place. Since we also observe the dot after Xist induction in undifferentiated ESCs, if the dot contains Xist transcripts these would originate from the endogenous Xist promoter of the castaneus X chromosome and not attributed to the doxycycline promoter on the TX chromosome. We believe that the low amount of Tsix and Xist do not materially affect our conclusions. In particular our new data on Xist induction by inducing differentiation in EpiLC medium suggest that neither the doxycycline promoter nor the genetic mutation of SPEN W522 affect the induction of Xist and observation of Xist clusters. We have added a brief statement that our probe also detects Tsix in the text on page 4.

It also seems that 4 days of dox induction of Xist expression is very long, as Xist is filling nearly the entire nucleus?

Response: We thank the reviewer and agree that the doxycycline induction system is fast and can initiate Xist expression very quickly. However, in our experiments mutations are analyzed that compromise Xist function in initiation of silencing and Polycomb modifications substantially. To discern the residual function of Xist we have used longer induction to observe chromatin modifications and gene silencing effects with greater sensitivity. We would also like to point out that our RNAseq data are obtained from total cellular RNA and represent steady state levels of transcripts that are not correlated to the transcription rate. To observe silencing we rely on turnover of the RNA. Four days of Xist induction ensures that a new steady state is reached where levels of transcripts in the cells correspond to the rate they are transcribed after gene silencing has been initiated.

Moreover, the images in Figure 4C suggest that the **Xist signal is more widely distributed** in the SPEN mutant line – is that the case? It seems the authors argue there is less Xist, but the image then is misleading.

Response: We have addressed the question raised by the reviewer by performing new quantitative measurements of the area of Xist signal in wild type and SPEN W522C carrying ESC lines and show the new analysis in new Fig. 5A and B. As the reviewer points out we observe a range of Xist cluster sizes that cover different areas. All clusters have a clearly identifiable intensity maximum around which signals become increasingly weaker. The maximal discernable area is about 30% of the nucleus in rare cases of wild type cells. We would like to point out that in these cases the distribution of signals might also include Xist molecules that might have been displaced from Xi as these appear sparsely distributed. In the case of cells carrying the SPEN W522C mutation focal Xist clusters are observed that are overall comparable to wild type cells. However, our quantitation shows that the area covered is slightly reduced, whereby the distribution and variability of cluster sizes between cells appears comparable. We discuss the localization of Xist in the W522C mutant cells in the revised text on page 6.

The **gene silencing data could be better presented with statistics and data quantification** (in Figure 4D and 4E). It is for instance unclear if there is any silencing in the SPEN mutant line. The data are somewhat undetailed.

Response: We have addressed the reviewer's concern by quantifying the average effect of Xist on X-linked genes and show the data with statistical significance in new Fig. 4E. For this we have generated new cell lines with deletions of the RRM region of SPEN as

controls. The cell lines are now included in the RNAseq data and calculation of silencing efficiency allowing us to observe that gene repression by Xist is substantially impaired by the SPEN W522C mutation but not entirely lost. We do see a complete absence of gene silencing in the SPEN mutant cell lines, but detect subtle reduction of X-linked gene repression in the SPEN W522C mutant cell lines.

The chromatin data in Figure 5 also lack any statistics and those for the CpG islands are particularly unclear.

Response: We have included additional experiments in our chromatin analysis in the revised version. Firstly, we include new cell lines with deletion of the RRM region of SPEN in our H3K27ac CUT and RUN analysis and H3K27me3 ChIP analysis. In addition we have extended the analysis of the W522C mutation by including a second replicate cell line. The new analysis is shown in revised Fig. 4F and G. We show transcription start sites and flanking regions for X-linked genes and autosomal genes as reference. This way the level of chromatin mark can be assessed in a qualitative manner. However, we would point out that relative quantification of chromatin marks beyond prominent shape changes of characteristic patterns i.e. peaks is difficult in our hands between different cell lines. Therefore, we draw our conclusions on principle and conservative observations from the chromatin data sets and refrain from quantitative statements. Irrespective of limitations, we observe a clear reduction of H3K27me3 recruitment to X-linked genes in cells carrying the W522C mutation of SPEN compared to wild type cells. Conversely, the deletion of SPEN RRM region prevents an Xist dependent increase in H3K27me3 over the transcription start site. We hope that the reviewer will agree that our qualitative statements are justified and add to our characterization of the consequences of mutating SPEN W522.

For the immunostaining of H3K27me3 and EZh2 not only the proportion of cells with enrichment should be considered but also the **intensity of the enrichment**.

Response: We thank the reviewer for the comment and agree that quantitation of immunofluorescence is difficult. Indeed, we do observe H3K27me3 foci in all cell lines, whereby both SPEN W522C as well as the RRM region deletion reduce the number of cells with H3K27me3 clusters. Since the appearance of clusters depends on the signal of the surrounding nuclear volume as well as the enrichment on Xi it is difficult to know what the threshold is for visually and subjectively observing a clear cluster. This assay works well when there is a large difference between control and mutant. However, it is

clear the SPEN mutations affect the efficiency of Polycomb recruitment but do not completely prevent it. For the reason that our observed differences between SPEN RRM deletion and W522C mutation are within experimental variability we have therefore removed the immunofluorescence staining data and refer to the CHIP data instead for making our conclusions in the revised version.

Minor Points:

At the bottom of page 23 and a few other places – are the authors referring to Table 3 rather than Table 4?

Response: We have corrected the mistake in the reference to Table 3 and thank the reviewer for pointing out the mistake.

Is there a table for the phosphoproteomics data that lead to the identification of S1284 phosphorylation?

Response: We have included a table with the identified phosphorylation sites of SPEN in the Supplemental Material of the revised version.

Response to Reviewer #4 Remarks to the Author

In this study the authors develop a CRISPR/Cas9-AID* Δ saturation mutagenesis strategy for analysing the function of individual genes of interest in mammalian cell culture models. They validate their system performing mutagenesis and 6TG selection for the X-linked Hprt gene in hemizygous XY ES cells and the MSH2 gene in haploid ES cells. They show that substitutions can be recovered for the majority of individual residues in both proteins, and that under selection, predominating mutations map to known functional domains and/or overlap with disease mutations seen in human disease attributable to the cognate genes. They go on to perform mutagenesis on the gene encoding the protein SPEN, a key factor for chromosome silencing in X chromosome inactivation. For this they make use of an established haploid XY ES cell line in which X inactivation can be induced using a doxycycline inducible Xist allele. Xist induction in these cells is ~ 90% cell lethal but SPEN loss of function rescues viability. Again, mutations are found across the length of this large (400kda protein), and following selection (Xist induction) several interesting substitutions are identified in SPEN functional domains, including the very extensive IDR regions of the protein. They

go on to show that one of the IDR mutations, S1284 is a site with high probability of being phosphorylated, and indeed confirm phosphorylation of this site specifically in cells with Xist RNA induction using phosphoproteomics. Finally, the authors investigate an additional function-linked SPEN mutation, W522, which occurs in the RNA binding RRM3-4 region of the protein in greater detail. They reproduce the W522 mutation in an XX mESC line to enable detailed analysis of X inactivation phenotypes. They show that W522 mutation results in a strong effect on SPEN function in gene silencing and in promoting histone deacetylation (H3K27Ac). They also observe decreased levels of Xist RNA in the SPEN mutant line, potentially linked to an RNA stability defect. They see no effect on the ability of Xist to recruit the PRC2 polycomb complex, contrasting with prior studies that suggested SPEN loss of function reduces or abrogates PRC2 and associated H3K27me3 in X inactivation. Based on this latter observation the authors suggest that W522 represents a separation of function mutation.

Overall this is an excellent and well conducted study. The strategy for saturation mutagenesis of genes of interest in mammalian cells represents a powerful tool that will be of interest across many different fields. Similarly, detailed functional maps of HPRT, MSH2 and SPEN provide a valuable resource for understanding how these proteins contribute to fundamental cellular processes and in human disease. There are nevertheless some points that need to be addressed, most notably in relation to the W522 mutation and how these findings should be interpreted in the context of previously published studies

Response: We thank the reviewer for the interest in our study and the helpful comments that have allowed us to further strengthen our conclusions.

1. It is not straightforward to reconcile how different studies have interpreted the effect of SPEN loss of function on Polycomb recruitment to Xi. Several different strategies have been applied to abrogate SPEN function, including RNAi knockdown, degron mediated acute depletion (with and without transgene complementation using full length or truncated constructs), in frame deletion of large domains of the protein, and engineered point mutations, notably of the SPOC domain. Added to this, different methods used to assay Polycomb on Xi range from IF for PRC2 and PRC1 subunits and/or of their cognate histone modifications (using various antibodies that likely have different avidity/affinity for epitope), and allelic ChIP-seq analysis, in most cases in interspecific XX mESC models. Moreover, different studies have used different timepoints of Xist expression either with or without differentiation, and different modes of Xist induction (native promoter vs TetOn system). The broad consensus from these studies is that SPEN mutations result in reduced but not loss of Polycomb on Xi. Both indirect (reduced gene silencing countering Polycomb deposition) and direct (for

example reduced levels/stability of Xist RNA) mechanisms could account for this (discussed also in <https://doi.org/10.1016/j.devcel.2021.04.009>). The authors of this study conclude that Polycomb recruitment is unaffected in W522 XX ES cells. However, the assays they use, scoring presence and absence of domains after IF, cannot be said to be very quantitative. The experiments lack a critical **positive control, XX mESCs with a SPEN loss of function mutation** that has been reported to reduce PRC2 recruitment, analysed with the same reagents and at the same induction timepoint as in this study (I actually couldn't find mention of how long Xist was induced in the experiments herein, only for the gene-silencing analysis. I assume the histone modification analysis was also with 4 days induction?).

Response: We thank the reviewer for the thoughtful comment and suggestion to compare the W522C mutation directly to a SPEN loss of function mutation. To address this request we have engineered two cell lines with homozygous deletions of the RRM region of SPEN during our revision. We use these cell lines along with a second replicate of a cell line carrying the W522 mutation in a new H3K27me3 ChIP analysis. The data are shown in new Fig. 4. We observe that a deletion of the SPEN RRM region prevents enrichment of H3K27me3 over the transcription start sites of X-linked genes after Xist induction. In contrast, after 4 days of Xist induction an increase of H3K27me3 over X-linked gene TSS can be observed for the SPEN W522C mutation. However, this increase is reduced compared to wild type cells. From this we conclude that SPEN W522C allows PRC2 marks to be established over genes downstream of Xist albeit at lower efficiency. In contrast, RRM deletions of SPEN do not result in such an increase after the same length of induction.

In addition to including an appropriate control, the authors should note that a prior study using ChIP-seq reported that Polycomb modification distribution across Xi is highly aberrant in SPEN null cells (thought to be due to aberrant Xist RNA localisation), but interestingly, not in cells with point mutations that abrogate SPEN-mediated gene silencing by blocking NCoR-HDAC3 interaction with the SPOC domain (doi: 10.1126/science.abe7500). This incidentally was also the first study to document that SPEN loss of function results in reduced Xist RNA stability and low Xist RNA levels and should be **referenced** in this context. Given that the authors have available interspecific XX mESCs with the W522 mutation, it would seem a good idea to perform a similar **ChIP-seq experiment to determine if distribution of Polycomb modifications (H3K27me3)** over Xi resembles that seen in wild-type (or SPEN SPOC mutant) or SPEN null cells. A further advantage of performing this experiment is it would allow for a more quantitative assessment of Polycomb deposition, eg using spike in or normalising to Xa signal.

Response: We thank the reviewer to pointing to the literature which we have included in the revised version. Our initial analysis had suggested that Xist localization as judged by RNA FISH was not materially changed by a SPEN W522C mutation. However, we have observed lower abundance in RT-PCR quantifications that might include also RNA that does not contribute to clusters. We have addressed the question of Xist localization raised by the reviewer by performing new quantitative measurements of the area of Xist signal in wild type and SPEN W522C carrying ESC lines and show the new analysis in new Fig. 5A and B. As the reviewer points out we observe a range of Xist cluster sizes that cover different areas. In our hands, all clusters have a clearly identifiable intensity maximum around which signals become increasingly weaker. The maximal discernable area is about 30% of the nucleus in rare cases of wild type cells. We would like to point out that in these cases the distribution of signals might also include Xist molecules that might have been displaced from Xi as these appear sparsely distributed. In the case of cells carrying the SPEN W522C mutation focal Xist clusters are observed that are overall comparable to wild type cells. However, our quantitation shows that the area covered is slightly reduced, whereby the distribution and variability of cluster sizes between cells appears comparable. These observations would indicate that indeed Xist RNA stability might be affected in SPEN mutant cells. However, the effect appears to be subtle and does not prevent the formation of clear clusters, which is similar to our previous observation of a SPEN RRM deletion, which showed clear Xist clusters but reduced recruitment of Polycomb marks (Monfort et al 2015). We have included a discussion of the effect of SPEN W522C on the localization of Xist and reference the study on Xist stability in the revised text on page 4. This study is consistent with an independent report of a requirement of SPEN for Tsix repression for Xist upregulation. We think that this effect might not apply to our Xist inducible system that uses a strong activation domain (the full VP16 transactivation domain) that has initially being selected for efficient Xist upregulation and likely overcomes Tsix and other repressors. However, at present we do not observe a strong effect of any of our SPEN mutations on Xist localization in this cell system or for the W522C mutation when X inactivation is initiated by cell differentiation in EpiLC medium (new Fig. 5C).

If as a result of these experiments it should emerge that W522 essentially phenocopies SPEN null mutations (as opposed to showing separation of function for the Polycomb phenotype) this would not in my view undermine the significance or value of this study. The availability of a range of specific point mutations affecting SPEN function will without doubt be of great value to the field as a whole.

Response: We thank the reviewer for the critical remark that has allowed us to substantially improve our experimental observations and correct a mistake in one of our earlier conclusions. During our revision we have isolated a second replicate for the W522C mutation of SPEN and compare two different cell lines to newly established cell lines with homozygous SPEN RRM deletions. This has allowed us to assess the dynamic range of our assays and also the reproducibility of the observations. The new data suggests that the W522C mutation is not a complete phenocopy of a null mutation of SPEN but retains a drastically reduced ability to support gene repression by Xist. This is consistent with our data on localization of a C-terminally RFP tagged SPEN W522C protein that shows drastically reduced localization and enrichment on Xi (new Fig. 5). At the same time H3K27me3 is established over the transcription start sites of X-linked genes. In our revised version we discuss now that it is conceivable that the reduced ability for gene silencing might be explained by a smaller fraction of X-linked genes being repressed and a considerable fraction remaining active. The enrichment of H3K27me3 might be explained from the fraction of genes that are silenced. Indeed, our new ChIP data suggest that the increase of the H3K27me3 signal over the transcription site is lower than in wild type controls consistent with the idea of recruitment to a smaller fraction of X-linked genes. This result also would appear in line with the literature that regards active transcription units exclude H3K27me3 modification.

2. The finding that S1284 phosphorylation occurs only in cells with Xist induction is curious given that SPEN likely functions at other sites as well (maybe not in ES cells?). Do the authors have a **preferred hypothesis for SPEN regulation by phosphorylation that they could add to the discussion?**

Response: We have investigated potential hints for the nature of the kinase but have not identified a link to Cdk8 or other candidate kinases with known functions in chromatin regulation. At the time the function of S1284 is not clear but an interesting focus of future investigation. Since S1284 has been identified in our screen as a high confidence mutation it is likely that a function for silencing depends on the serine at position 1284 as the picked up point mutations seem not to generate new cleavage sites or other signals that would have detrimental effects on protein stability or localization. However, such possibilities cannot be ruled out as computational analysis might be limited. The Xist dependent gain of phosphorylation could be explained as a consequence of localizing to the Xi where specific kinases might be activated. Predictions for kinases would likely involve cyclin dependent kinases from Cdk8 and the cyclin binding protein CIZ1 have been implicated in X inactivation. However, S1284 does not appear a target of the class of cyclin dependent kinases, which makes further predictions speculative as is the function of phosphorylation of S1284 for silencing.

3. Decreased levels of Xist RNA in SPEN W522 mutant mESCs is shown only as shades of green on a heat map (as far as I could find) and I think it would be useful to have a **more quantitative measure, eg qRT-PCR**.

Response: We show the level of Xist expression in new Fig. 4D. These are TPM normalized read counts from our RNAseq datasets. The data show that Xist is induced from the doxycycline inducible promoter in all cell lines but reaches the highest level in wild type cells. As reference we show also the level of Xist induction after entry into differentiation by a shift to EpiLC medium, which is comparable to the levels seen with the doxycycline inducible promoter. We need to point out that the levels are not identical and span a 3-fold range. Our suggestion is that TPM values include reads of Xist RNA that has been displaced from the Xi and might be in the process of being degraded. This might point to a considerable overproduction of Xist from the doxycycline inducible promoter as the levels observed after entry into differentiation appear nearly identical (Fig. 5D right columns).

POINT TO POINT RESPONSE TO THE REVIEWERS' COMMENTS

Response to Reviewer #1:

I am satisfied with the revisions that have been made by the authors.

Response: We thank the reviewer for supporting publication of our study.

Response to Reviewer #2:

I appreciate the extensive edits and additions made by the authors and am satisfied with them. They help to better understand the impact of the W522 mutation in SPEN.

Response: We thank the reviewer for the helpful comment and supporting publication of our study.

Response to Reviewer #4:

The authors have included new data and analysis RE SPEN mutations, and have revised conclusions somewhat, largely addressing the issues that I raised in my original review. Some points are not fully resolved but addition of some caveats and a couple of minor corrections should be sufficient to deal with these;

1. Introduction lines 34 and 35 do not capture all points from literature on SPEN effects on polycomb. As I remember the two referenced papers looked only at H3K27me3 so generalising with the term Polycomb is not factually correct. Reference 51 reported on both H3K27me3 and H2AK119ub1 in SPEN RRM and SPOC mutations, observing that both modifications are enriched on Xi in the RRM and SPOC mutation lines but that the RRM mutation affects distribution of both modifications on Xi, implying an effect on where Xist localises (see also below).

Response: We thank the reviewer for the thoughtful comments and support of publication of our study. In response, we have rewritten the text to include the suggested changes. In response to point 1, the two studies cited in the paragraph starting line 34 contain immunofluorescence experiments showing changes in localization of the Polycomb proteins Ezh2 and Ring1b as well as H3K27me3 in Spen mutant cells after Xist induction. We refer to this immunofluorescence data as basis of our statement as the observed effects show that loss of SPEN has a dramatic effect on the ability of Xist to cause an enrichment of these two Polycomb proteins or H3K27me3. Reference 51 uses super resolution microscopy to assess Polycomb localization, which is consistent with earlier immunofluorescence staining but reveals additional detail. We cite reference 51 in the text.

2. Results, line 172 states 'exhibited slight X chromosomal gene expression...' – shouldn't this be other way round, ie. slight gene repression?

Response: We have corrected the mistake and rephrased the sentence.

3. Line 177 mentions lower Xist levels in W522C/delta but isn't this also the case for W522C/Ins?

Response: We have added W522/Ins to the sentences as the reviewer correctly observed that Xist levels are reduced in both mutant cell lines.

4. Line 187-190. I didn't understand why the authors confined their H3K27me analysis to transcription units given gain of polycomb modifications on Xi occurs widely over CREs and intergenic regions. As noted in the rebuttal letter loss of H3K27me3 over transcription units is expected to correlate with the transcription that occur on the inactive X in SPEN mutants. Also reference 51 observed changed distribution of polycomb modifications in SPEN RRM but not SPOC mutants. As a minimum this section needs to be more carefully worded to state 'H3K27me3 over transcription units'.

Response: We have amended the sentence to indicate that we refer to X-linked transcription units when discussing our H3K27me3 ChIP data in the text. The transcription units are the relevant aspect as non-genic Polycomb recruitment has been shown to be independent of Xist repeat A, which is thought to act through SPEN recruitment. Therefore, non-genic H3K27me3 would not be expected to be directly affected in SPEN mutant cells. Hence, our focus on the genic H3K27me3 through analysis of transcription units.

5. Line 198. The FISH analysis showing clusters are unaffected in SPEN mutations needs to be interpreted cautiously as it does not assess where Xist localises and there are caveats in the methodology where sparsely distributed molecules in mutant cells may artificially increase apparent size of clusters. The authors acknowledge this point in rebuttal but it seems to be lost in the manuscript. As a minimum a caveat to say that whilst Xist localisation appears normal in the SPEN mutants, the assays used do not assess whether distribution over the chromosome changes.

Response: We have added a statement following the reviewer's request for pointing out that our analysis has been performed at a lower resolution than the super resolution microscopy in reference 51, which might limit our ability to observe smaller changes in Xist localization.

6. Consider swapping the position of the two paragraphs in discussion as would seem more logical to first discuss the findings relating to saturation screening and then finishing with the discussion of SPEN analysis?

Response: We thank the reviewer for the suggestion. We have considered the suggested change in ordering the paragraphs of our discussion. However, we feel that an ending in a general outlook would better orient the reader towards future use of our methodology. Hence, we prefer to maintain the opening with discussion of our conclusions of SPEN function in X inactivation.

7. Line 689 figure legends, missing H3K27me3 in heading.

Response: We thank the reviewer for pointing out the missing identification and have added H3K27me3 to the legend of figure 4G.